# Analysis of gut microbiota in patients with AVS and identification of potential biomarkers: a cross-sectional study

Fei Jiang,[1,2,3] Meiling Cai,[1,2,3] Yanchun Peng,[1,2] Sailan Li,[1,2] Yuling Xie,[2,3] Qiong Pan,[1,2] Jianlong Lin,[1,2] Bing Liang,[1,4] Liangwan Chen,[2,3] Yanjuan Lin[1,2,3]

**ABSTRACT** This study evaluated the characteristics of the gut microbial (GM) in patients with aortic valve stenosis (AVS). Thirty patients diagnosed with AVS and 30 healthy controls (HC) were included. Fecal samples were obtained for high-throughput 16S rDNA sequencing. Bacterial diversity was assessed using QIIME and R software. Potential biomarkers were identified using the random forest model. The model performance was evaluated using receiver operating characteristic (ROC) curves and decision curve analysis. Relationships between the GM and clinical characteristics of participants were examined using the Spearman's correlation. The composition of GM, measured by beta diversity, significantly differed between the two groups (Adonis, $P = 0.001$), indicating distinct microbial community structures in AVS patients compared to HC. At the phylum level, the *Firmicutes/Bacteroidetes* (F/B) ratio was significantly lower in the AVS group compared to the HC group ($P = 0.031$, Wilcoxon test). At the genus level, the relative abundance of short-chain fatty acids (SCFAs)-producing bacteria, including *Lachnospiraceae*, *Prevotellaceae*, and *Enterococcus,* was significantly reduced. Twenty-four genera were identified as potential biomarkers using the nested cross-validation feature of the random forest model (86.67% accuracy in cross-validation). The area under the receiver operating characteristic curve (AUC) was 0.94 (95% CI = 0.79–1.00). The decision curve analysis highlighted the practical utility of the model in clinical settings. Patients with AVS exhibited alterations in GM, particularly a reduction in the SCFAs-producing bacteria. The distinct GM profiles demonstrated strong predictive capabilities for AVS and were related to the worsening of clinical indicators. Notably, 24 genera may serve as potential biomarkers and predictors in clinical settings.

**IMPORTANCE** Dysbiosis of GM contributes to cardiovascular diseases; however, research on GM alterations in patients with aortic valve stenosis (AVS) is limited. The study aimed to conduct a cross-sectional study matched for age, sex, body mass index (BMI), and patient geographic region, to analyze the GM in patients with AVS, to identify potential biomarkers, and to assess the effectiveness of clinical prediction. If this correlation is confirmed, the GM may be used for risk stratification and identification of potential therapeutic targets.

**CLINICAL TRIALS** This clinical study was registered with the China Clinical Trials Registration Center (No. ChiCTR2400081198).

**KEYWORDS** aortic valve stenosis, gut microbiota, short-chain fatty acids, random forest

Aortic valve stenosis (AVS) prevalence increases with age, making it the third most common cardiovascular disease in older adults (1, 2), affecting approximately 12.4% of the population. By 2040, the prevalence is expected to increase by 2.4 times globally (3–5). Severe AVS, if not treated promptly, has a median survival of only 1.8 years, with

**Peer Reviewer** Djandan Tadum Arthur Vithran, Central South University, Changsha, China

Address correspondence to Liangwan Chen, fjxhlwc@163.com, or Yanjuan Lin, fjxhyjl@163.com.

Fei Jiang and Meiling Cai contributed equally to this article. The author order was determined both alphabetically and in order of increasing seniority

The authors declare no conflict of interest.

See the funding table on p. 12.

a high mortality rate of 50% within 2 years (6, 7). Three-quarters of patients progress to heart failure or mortality within 5 years (5). This represents a significant threat to public health. Surgery remains the only treatment option for improving survival for patients with severe AVS (8). Despite recent advancements in interventional therapies, the incidence and mortality rates of AVS remain high. Therefore, this study aims to investigate early biomarkers and potential intervention targets as crucial steps in developing new strategies for AVS.

The exact pathogenic mechanism of AVS remains unclear; however, inflammation is considered a key factor facilitating its progression. Recent research suggests that GM may influence the occurrence and development of cardiovascular diseases by regulating inflammatory cell levels through the gut–heart axis (9–11). Therefore, focusing on the GM could offer new therapeutic opportunities for preventing and treating AVS. Currently, there is limited research on the role of GM in patients with AVS. Previous studies have suggested that AVS and atherosclerosis (AS) share common clinical risk factors (12, 13). Currently, there is ongoing controversy regarding GM differences in patients with AS. Sawicka-Śmiarowska et al. demonstrated significant variations in both α- and β-diversity of GM between patients with AS and HC (14). In contrast, Dong et al. (15) reported divergent findings: while no significant difference was observed in α-diversity, β-diversity exhibited notable variations associated with the enrichment of inflammation-related microbial taxa. Liu (16) discovered similar findings in the GM of patients with calcific aortic valve disease, where no alpha diversity was observed. However, the composition of bacterial communities in the two groups did not overlap. Significant age variation existed between the two study groups, but the proportion specific to AVS was not clarified. Additionally, Kocyigit et al. (17) identified differences in gut microbiota-related metabolite levels in the plasma of patients with calcific AVS, which correlated with worsening clinical indicators. However, the study did not explore the differences in GM. These studies highlight the complex and sometimes contradictory role of GM in cardiovascular diseases with similar risk factors, suggesting both protective and pro-inflammatory roles depending on the disease context. Currently, no domestic or international studies clearly report GM diversity in patients with AVS, and the relationship with the severity of the disease is also unknown. The presence of characteristic bacterial profiles in AVS requires further investigation. Therefore, we aimed to conduct a cross-sectional study matched for age, sex, BMI, and geographic region. We analyzed the GM in patients with AVS, identified potential biomarkers, developed a random forest model to classify the disease based on these biomarkers, and assessed the effectiveness of clinical prediction. Additionally, we analyzed the relationships between GM and patients' clinical characteristics. The identification of GM profiles in AVS may not only serve as potential biomarkers but also guide future therapeutic interventions targeting the gut–heart axis in cardiovascular diseases.

## MATERIALS AND METHODS

### Study participants

A cross-sectional study was conducted at the Fujian Heart Medical Center between February and May 2024. The study included 30 patients with AVS and 30 HC, matched for sex, age, BMI, and geographical region, recruited from the physical examination center during the same period. The inclusion criteria were as follows: (1) age ≥18 years (2) and patients with severe AVS diagnosed using cardiac ultrasound assessment with a valve opening area ≤1.0 cm$^2$ and/or peak aortic gradient ≥50 mmHg. The exclusion criteria were as follows: (1) use of probiotics, glucocorticoids, or antibiotics within 3 months before stool sampling (2); history of intestinal surgery (3); episodes of recurrent diarrhea or constipation within the past month; and (4) presence of digestive system tumors, digestive tract infections, inflammatory bowel disease, or other related conditions.

## General demographic information and clinical characteristics

A custom questionnaire was used to collect general demographic information and clinical characteristics of the participants, such as sex, age, BMI, and history of diabetes and hypertension. Data on monocytes, white blood cells (WBCs), lymphocytes, neutrophils, brain natriuretic peptide (BNP), aortic valve orifice area (AVOA), peak aortic gradient (PAG), mean aortic gradient (MAG), cardiac output (CO), cardiac index (CI), and length of stay (LOS) were retrieved from medical records.

## Fecal sample collection and DNA extraction

Upon admission, participants were trained in fecal sampling techniques and given sterile fecal sampling kits. They were instructed to avoid fecal contamination with urine and to exclude diarrhea or loose stools during sampling. To minimize the confounding effects of diet and hospital environment exposure, a sterile cotton swab was used to collect fecal material from multiple sites within the sample, following a standardized "#" pattern to reduce sampling bias and capture the diversity of the microbiota. Fecal samples from patients with AVS were collected within 24 h of admission, whereas healthy individuals at the physical examination center of the hospital voluntarily provided their fecal samples as controls. The fecal samples were stored at $-20°C$ within 15 min of collection, transferred to our laboratory on dry ice within 2 h, and stored at $-80°C$. The fecal samples, packed with dry ice, were sent for analyses to the laboratory at LC-Bio Technology Co., Ltd., Hang Zhou, Zhejiang Province, China.

## DNA extraction and sequencing

Total fecal microbial DNA was extracted using the Fecal Genome DNA Extraction Kit (AU46111-96, BioTeke, China) by LC-Bio, following ISO 9001-certified protocols and strictly adhering to the manufacturer's instructions. The DNA was quantified using Qubit (Invitrogen, USA). The process was strictly performed according to the manufacturer's protocol. Total DNA was amplified by PCR using the universal primer 341F/805R(341F: 5'-CCTACGGGNGGCWGCAG-3'; 805R:5'-GACTACHVGGGTATCTAATCC-3'). The PCR amplification conditions were: pre-denaturation at 98°C for 30 s; denaturation at 98°C for 10 s; annealing at 54°C for 30 s; extension at 72°C for 45 s; for a total of 32 cycles. The final extension was at 72°C for 10 min. The PCR product was purified using AMPure XP Beads (Beckman Coulter Genomics, Danvers, MA, USA) and quantified using Qubit (Invitrogen, USA). All 60 samples were processed in a single batch using identical library preparation protocols and sequenced in one Illumina NovaSeq 6000 to eliminate technical variability. Qualified PCR products were evaluated using an Agilent 2100 Bioanalyzer (Agilent, USA) and Illumina library quantitative kits (Kapa Biosciences, Woburn, MA, USA). The libraries were then pooled together and sequenced on an Illumina NovaSeq 6000 (PE250), provided by LC-Bio Technology Co., Ltd., Hangzhou, China.

## Data processing and 16S rDNA sequencing analysis

As previously reported (18), the sequencing primer was removed from the de-multiplexed raw sequences using Cutadapt (v 1.9). Then, paired-end reads were merged using FLASH (v 1.2.8). The low-quality reads (quality scores <20), short reads (<100 bp), and reads containing more than 5% "N" records were trimmed using the sliding-window algorithm method in fqtrim (v 0.94). Quality filtering was performed to obtain high-quality clean tags according to fqtrim. Chimeric sequences were filtered using Vsearch software (v2.3.4). DADA2 was applied for denoising and generating amplicon sequence variants (ASVs). ASVs using QIIME2's DADA2 pipeline, with a 99% similarity threshold, as opposed to the traditional 97% threshold, were used for operational taxonomic units (OTUs). The sequence alignment of species annotation was performed by the QIIME2 plugin feature-classifier, and the alignment databases used were SILVA and NT-16S. Alpha and beta diversities were calculated using QIIME2. Relative abundance was used

in bacteria taxonomy. The Wilcoxon test was used to identify the differentially abundant genera, and significances were declared at $P < 0.05$. LDA effect size (LEfSe, LDA ≥ 3.0, $P$-value < 0.05) was performed using nsegata-lefse. Based on the full-length 16S rDNA gene sequence database from Greengenes, the PICRUSt (Phylogenetic Investigation of Communities by Reconstruction of Unobserved States) method was used to predict microbial community functions by comparing with the Kyoto Encyclopedia of Genes and Genomes (KEGG) database. Based on the metabolic pathways in the KEGG database (KEGG pathway), biological metabolic pathways can be classified into three levels. The first-level categories include cellular processes, organismal systems, genetic information processing, environmental information processing, metabolism, human diseases, and drug development. Other diagrams were implemented using the R package (v 3.4.4).

## Statistical analyses

Statistical analyses were conducted using IBM SPSS Statistics for Windows 26.0 IBM (Corp., Armonk, N.Y., USA) and R v 3.5.1 (R Foundation for Statistical Computing, Vienna, Austria). The Shapiro-Wilk test was deployed to determine if the continuous variables conformed to a normal distribution. Normally distributed data are presented as the mean ± SD, and Student $t$-tests were employed to conduct comparisons between groups. For non-normally distributed variables, data were presented as the median (interquartile range), with comparisons conducted utilizing the nonparametric Wilcoxon test. Categorical data were presented as frequencies and percentages, with comparisons made using the $\chi^2$ test or Fisher's exact test, as appropriate. Random forest models were trained using the R Random Forest package to predict AVS. A probability matrix generated during the random forest procedure and a sample metadata file were employed to create the decision curve analysis (DCA) in R. Spearman's rank correlation was utilized to assess the relationships among non-normally distributed data. $P$-values were adjusted for multiple comparisons using the Benjamini-Hochberg method, which controls the false discovery rate, ensuring more robust conclusions from the analyses. A $P$-value < 0.05 was considered statistically significant.

## RESULTS

### General data and laboratory indicators

Overall, 30 patients with AVS and 30 HC were included in the AVS and HC groups, respectively. No significant differences were found between the two groups in terms of age, BMI, sex, history of hypertension, history of diabetes, and dietary habit ($P > 0.05$). However, the levels of WBCs, monocytes, and other related inflammatory factors were significantly higher in the AVS than in the HC group ($P < 0.05$) (Table 1).

### Comparison of gut microbiota diversity between the AVS and HC groups

16S rDNA sequencing was completed in June 2024 for the 30 stool samples collected from patients with AVS and 30 from HC. Using the ASV approach, OTU analogs were constructed, and an ASV feature table and characteristic sequences were obtained (Table S1). For alpha diversity analysis, 35,745 sequences were randomly selected from each sample based on the lowest sequence count. The dilution curve was generated using the following parameters: --p-max-depth 35,745--p-min-depth 1 --p-steps 10 --p-iterations 10. Both the Chao1 and Shannon index dilution curves reached a saturation point, with coverage values ranging from 0.9994 to 0.9998. This indicates that the sequencing depth was adequate to capture the microbial diversity (Shannon index) and richness (Chao 1 index) of the samples (Fig. 1A). Microbial alpha diversity analysis for each participant revealed no significant differences in diversity or richness between the two groups (Chao 1 index, $P = 0.37$; Shannon index, $P = 0.49$) (Fig. 1B). PCoA analysis based on the unweighted_unifrac dissimilarity showed distinct variation in GM composition between patients with AVS and HC (beta diversity, Adonis, $R^2 = 0.06$, $P = 0.001$, Fig. 1C). The GM of the AVS and HC groups contained 2,127 ASVs, of which 1,253 were shared between

**TABLE 1** Demographic and clinical characteristics of subjects[a]

| Characteristic | AVS group (N = 30) | HC group (N = 30) | t/$\chi^2$ | P value |
|---|---|---|---|---|
| Age, years | 66 (59, 74) | 66 (62, 71) | 0.404 | 0.594 |
| Male | 17 (56.67) | 17 (56.67) | 0.00 | >0.999 |
| BMI (kg/m$^2$) | 22.67 (20.57, 24.44) | 23.68 (22.99, 25.22) | −1.722 | 0.086 |
| Dietary habit | | | 0.351 | 0.554 |
| Eastern diet | 28 (93.3) | 29 (96.7) | | |
| Western diet | 2 (6.7) | 1 (3.3) | | |
| Mediterranean diet | 0 (0) | 0 (0) | | |
| History of diabetes | 4 (13.33) | 5 (16.67) | 0.131 | 0.718 |
| Hypertension | 17 (56.67) | 15 (50.00) | 0.268 | 0.605 |
| WBCs 10$^9$/L | 5.90 (5.40, 7.20) | 5.25 (4.31, 5.97) | 2.484 | 0.013 |
| Neutrophils, 10$^9$/L | 3.45 (2.66, 3.93) | 3.03 (2.71, 3.45) | 1.212 | 0.228 |
| Lymphocytes, 10$^9$/L | 1.56 (1.24, 2.29) | 1.92 (1.44, 2.22) | 1.3271 | 0.206 |
| Monocytes, 10$^9$/L | 0.45 ± 0.15 | 0.36 ± 0.12 | 2.550 | 0.013 |
| BNP, pg/mL | 667.00 (149.75, 2099.25) | | | |
| AVOA, cm$^2$ | 0.78 (0.63, 0.90) | | | |
| PAG, mm Hg | 91.50 (86.00, 101.75) | | | |
| MAG, mm Hg | 57.00 (52.00, 67.25) | | | |
| LOS, day | 11.00 (8.00, 18.25) | | | |
| CO, L/min | 4.85 (3.80, 6.28) | | | |
| CI, L/min.m$^2$ | 2.90 (2.45, 3.80) | | | |

[a]BMI, body mass index; WBCs, white blood cell; BNP, brain natriuretic peptide; AVOA, aortic valve orifice area; PAG, peak aortic gradient; MAG, mean aortic gradient; LOS, length of stay; CO, cardiac output; CI, cardiac index.

both groups. Among these, 471 ASVs were unique to the AVS group, whereas 403 ASVs were distinct to the HC group (Fig. 1D). Significant variations in the GM were identified between the two groups that met the criteria for cluster analysis. A phylogenetic tree constructed using ASVs demonstrated that these ASVs belonged to five main phyla: *Firmicutes*, *Bacteroidetes*, *Actinobacteria*, *Proteobacteria*, and *Verrucomicrobia* (Fig. 1E).

## Taxonomic analysis of the GM composition of the AVS and HC groups

At the phylum level (Fig. 2A), the predominant fecal microorganisms included *Bacteroidetes*, *Firmicutes*, *Actinobacteria*, and *Proteobacteria*. The value of F/B is generally considered an indicative of GM imbalance (19). We found that the *Firmicutes/Bacteroidetes* (F/B) ratio was significantly higher in the HC than in the AVS group (P = 0.031, Wilcoxon test). At the genus level (Fig. 2B), the most prevalent bacteria were *Bacteroides*, *Enterococcus*, *Escherichia-Shigella*, *Faecalibacterium*, and *Streptococcus*. The evolutionary cladogram and histogram (Fig. 2C and D) depict taxonomic levels from phylum to species as concentric circles, with the diameter of each circle proportional to the relative abundance in the GM. The letters p, c, o, f, g, and s correspond to the phylum, class, order, family, genus, and species, respectively. LEfSe analysis performed across taxonomic levels from phylum to species identified bacterial genera with significant variations between the AVS and HC groups. The results showed that 19 bacterial taxa were enriched in the HC group, whereas 31 were in the AVS group. Patients with AVS displayed lower abundances of *Lachnospiraceae* (P < 0.0001, FDR corrected q < 0.0001), *Akkermansia* (P = 0.0003, FDR corrected q = 0.02), *Enterococcus* (P = 0.0002, FDR corrected q = 0.01), Clostridiales_unclassified (P = 0.0003, FDR corrected q = 0.01), and *Prevotellaceae* (P = 0.0011, FDR corrected q = 0.03). In contrast, patients with AVS had higher abundances of *Megamonas* (P = 0.0072, FDR corrected q = 0.2098).

## Functional gene analysis in the KEGG database

From Table S2, it is evident that genes encoded by gut bacteria are involved in various functions. At the third-level classification, compared to the normal control group, the gut

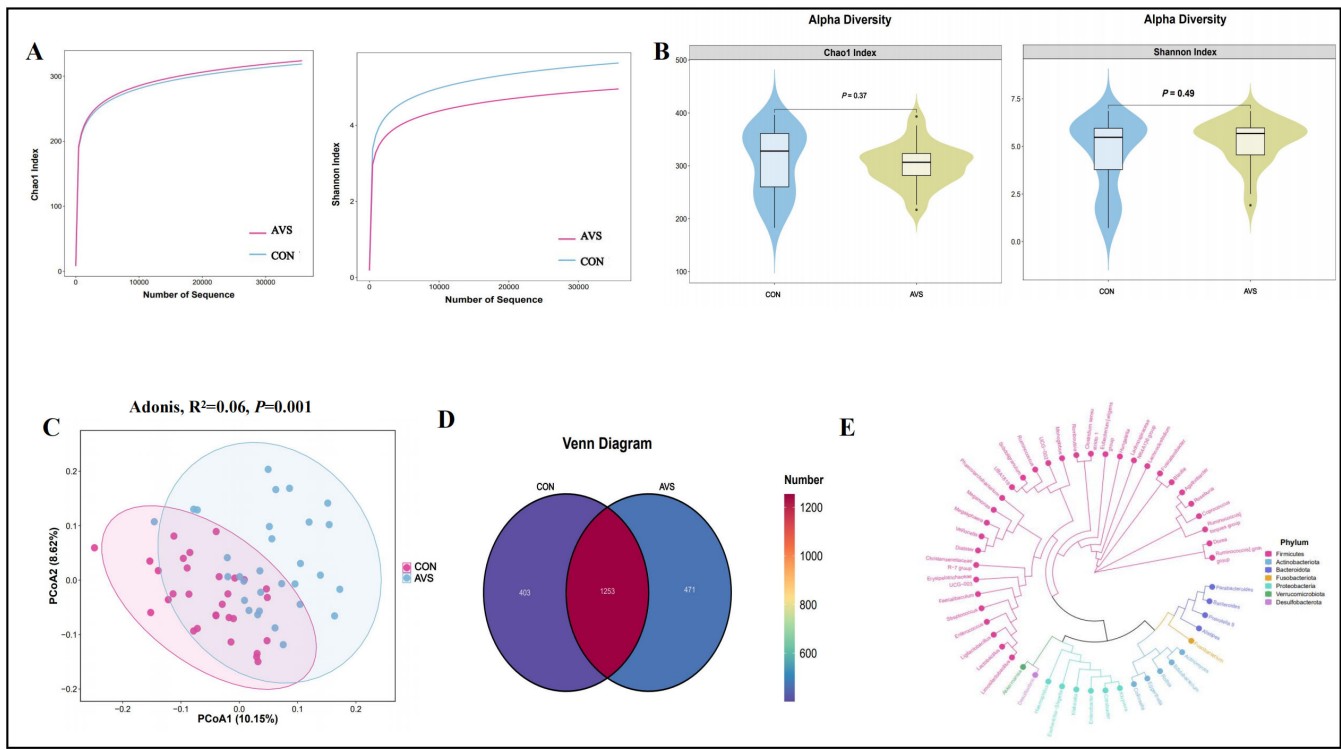

**FIG 1** Diversity of the GM in patients with AVS and HC. (A) Rarefaction curves for Chao 1 and Shannon indices. (B) Comparison of alpha diversity, as indicated by the Chao 1 and Shannon indices between the two groups. No significant differences were observed in alpha diversity between the AVS and HC groups (Chao 1 index, $P = 0.37$; Shannon index, $P = 0.49$, Wilcoxon test). (C) Ordination plot showing the first two PCoA based on unweighted_unifrac dissimilarity. Samples from the AVS and HC groups clustered relatively closely, indicating that unweighted_unifrac dissimilarity differs between the groups (Adonis, $P = 0.001$). (D) Venn diagram showing the number of unique and shared ASVs between the two groups. (E) A phylogenetic tree was constructed using ASVs, with ASVs color-coded by phylum. ASVs, amplicon sequence variants; PCoA, principal coordinate analysis; Adonis, non-parametric multivariate analysis of variance; $N = 30{:}30$.

microbiota in the AVS group exhibited changes in several functions. Specifically, the AVS group showed a decrease in the functions of glycolysis/gluconeogenesis, transporters, phosphotransferase system, and dioxin degradation ($P < 0.05$), while functions such as porphyrin and chlorophyll metabolism, histidine metabolism, arginine and proline metabolism, oxidative phosphorylation, energy metabolism, chaperones and folding catalysts, and biotin metabolism were enhanced ($P < 0.05$), as shown in Fig. 3.

## Twenty-four genera may potentially serve as biomarkers in AVS diagnosis

To evaluate whether GM could distinguish patients with AVS from healthy controls (HC), we applied a random forest model at the genus level. The model achieved 86.67% accuracy and an AUC of 0.94 (95% CI = 0.79–1.00), demonstrating strong discriminatory power. We then identified the most relevant bacterial taxa associated with AVS. Using tenfold nested cross-validation, we found that the out-of-bag error rate decreased as the number of genera increased but plateaued beyond 24 genera (Fig. 4A), indicating 24 as the optimal biomarker count. Fig. 4B highlights the top 24 genera by Gini index, including *Lachnospiraceae*, *Akkermansia*, and *Enterococcus*—consistent with LEfSe results. Further validation via leave-one-out cross-validation confirmed the model's robustness (AUC = 0.94; Fig. 4C). Decision curve analysis (DCA) revealed that the model provided greater net benefit than the "all" or "none" strategies at threshold probabilities of 0.19–0.96 (Fig. 4D).

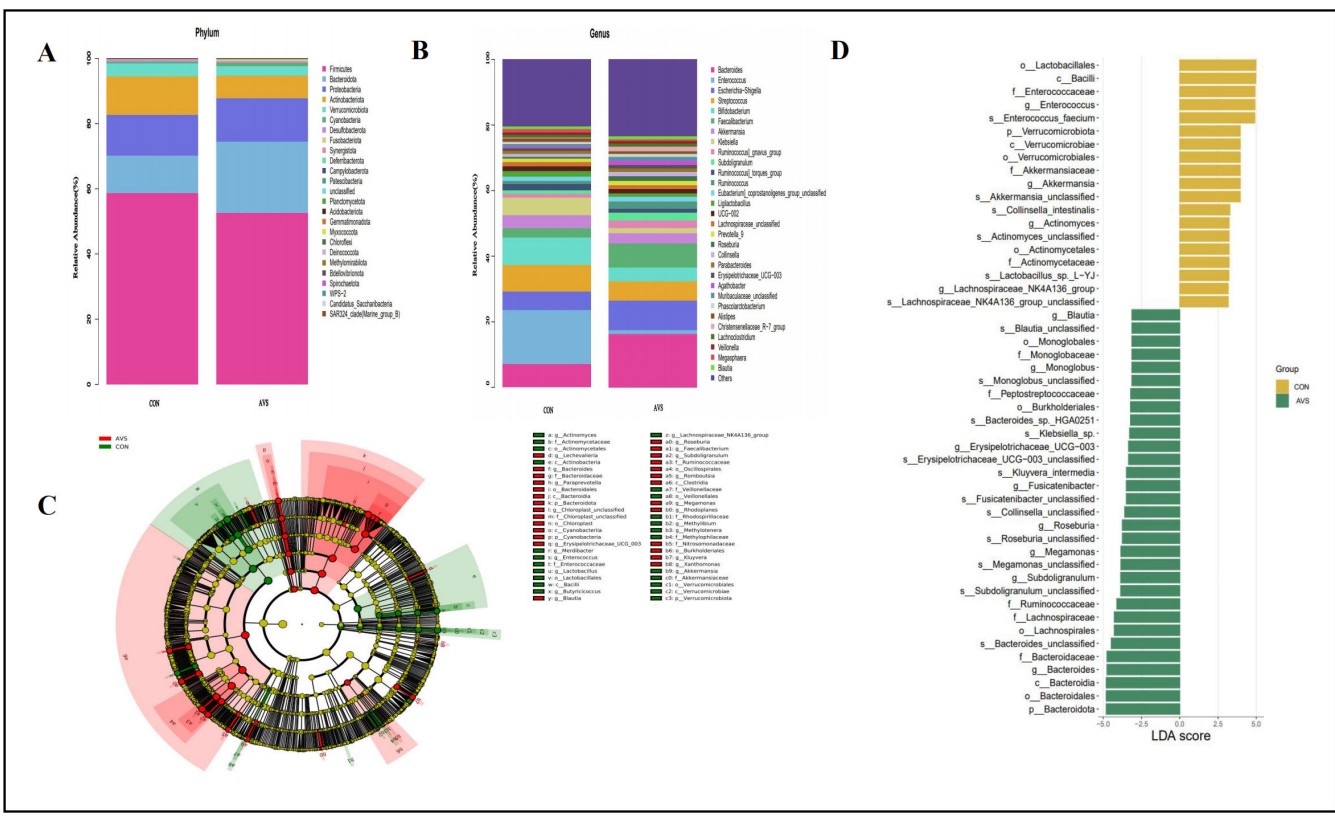

**FIG 2** Taxa differences between patients with AVS and HC. Relative abundances of the major bacteria at the (A) phylum and (B) genus levels. (C) Cladogram illustrating the phylogenetic distribution of the differential GM between the two groups. The radiating structure from the center to the outer edge represents seven taxonomic levels, with each node corresponding to a species classification at that level. Larger nodes indicated higher species abundance. Yellow nodes show no significant differences between the groups, whereas red nodes represent species with significantly higher abundance in the AVS group. The other colors indicate varying levels of abundance between the groups. (D) Differences in intestinal bacteria between the two groups. LefSe analysis (LDA score >3) was used to identify major differences in bacterial taxa. Green bars indicate genera with higher relative abundances in patients with AVS, whereas yellow bars indicate genera that were more abundant in the HC group. The x-axis represents the log LDA scores. LDA, linear discriminant analysis; $N$ = 30:30.

## Relationships between the GM and clinical characteristics

To investigate the relationship between the GM and clinical characteristics, Spearman's correlation analysis was conducted among the top 24 differential bacterial genera and the relevant clinical features of the participants. Heatmap analysis (Fig. 5A) showed that bacterial genera with lower abundances in the AVS group, compared to the HC group, were positively correlated with inflammatory markers. These genera were negatively related to disease severity (AVOA and mean cross-valve pressure). Scatter plots for the genera with decreased abundance of intestinal flora (Fig. 5B through D) demonstrated that the abundance of *Lachnospiraceae* negatively correlated with WBC counts ($R$ = −0.528, $P$ < 0.0001), negatively correlated with the Monocyte ($R$ = −0.699, $P$ < 0.0001), and positively correlated with AVOA ($R$ = 0.786, $P$ < 0.0001).

## DISCUSSION

In this study, sequencing of the V3–V4 region of the 16S rDNA gene was performed using high-throughput techniques to compare the GM characteristics of HC and patients with AVS. To our knowledge, this is the first study to investigate the distinctive microbiota profile of AVS by matching BMI, sex, age, and geographic region. The study found high similarity in GM uniformity and richness between the two groups. However, beta diversity analysis revealed significant differences in GM composition between the groups. Compared with that of the control group, F/B ($P$ = 0.031, Wilcoxon test) in the

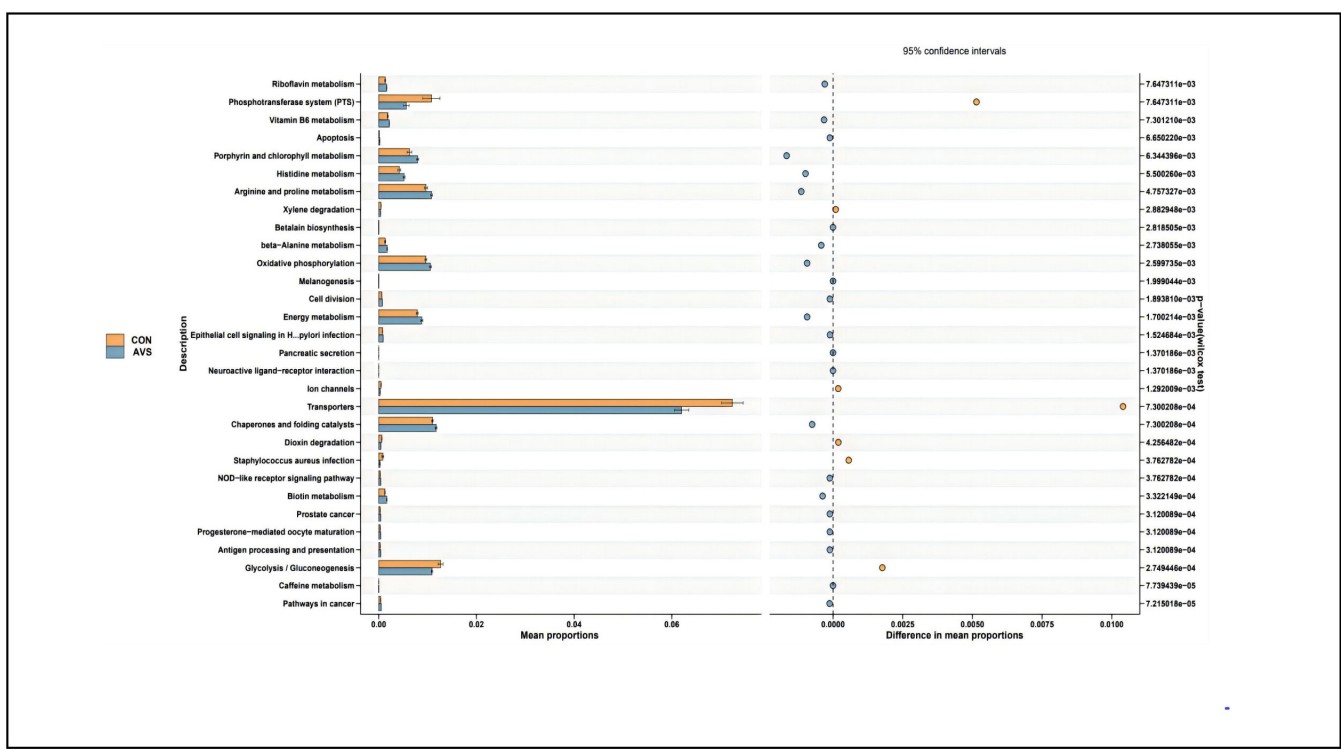

FIG 3 Functional enrichment of KEGG at level 3. The horizontal coordinate on the left of the picture shows the proportion of differential functions. The taller the column, the higher the proportion, and the function of the vertical coordinate is displayed from bottom to top according to the *P*-value from small to large. The bubble chart on the right shows the abundance difference of each function. The group in which the difference function is enriched can be judged by the color of the points. The distance between the points and the dashed line in the middle is the abundance difference of the two groups of functions. The difference analysis was performed using the Wilcoxon test (wilcox.test), *N* = 30:30.

AVS group decreased at the phylum level. At the genus level, the relative abundance of SCFA-producing bacteria, including *Lachnospiraceae*, *Prevotellaceae*, *Lactobacillus*, and *Enterococcus*, decreased. Using nested cross-validation function of the random forest, 24 genera were identified as potential biomarkers. A random forest model constructed with these genera was used to distinguish patients with AVS from HC with an accuracy of 86.67% and an area under the receiver operating characteristic curve of 0.94. Decision curve analysis confirmed that the model holds practical value in clinical applications.

At the phylum level, the GM was composed of five phyla: *Bacteroidetes*, *Firmicutes*, *Actinobacteria*, *Proteobacteria*, and *Verrucomicrobiota*. In healthy individuals, *Bacteroidetes* and *Firmicutes* together constituted more than 90% of the GM bacterial species, and a change value of F/B is generally considered a sign of GM imbalance (20). Compared with that of the HC groups, patients with AVS exhibited a significantly lower F/B ratio, which aligns with the findings by Emoto (21). Previous research suggests that *Firmicutes* are beneficial bacteria, which produce more butyrate (22). SCFAs are widely recognized for their health-promoting effect. Butyrate enhances insulin sensitivity and exerts an anti-inflammatory effect (23). In contrast, Bacteroidetes primarily produce propionate, which can induce inflammation by regulating the PTEN/AKT pathway (24). Therefore, we hypothesize that the decreased F/B ratio reflects a shift from anti-inflammatory bacteria (e.g., *Firmicutes*) to pro-inflammatory bacteria (e.g., *Bacteroidetes*), which may contribute to the inflammatory processes observed in the AVS group. This finding is consistent with the increased inflammatory markers (WBCs and monocytes), as shown in Table 1 and Fig. 5.

At the genus level, further LEfSe analysis revealed decreased abundance of *Lachnospiraceae*, *Prevotellaceae*, *Enterococcus*, *Lactobacillus*, *Eubacterium*, and *Akkermansia* in the AVS group. Significantly, all of these genera, except *Akkermansia*, are

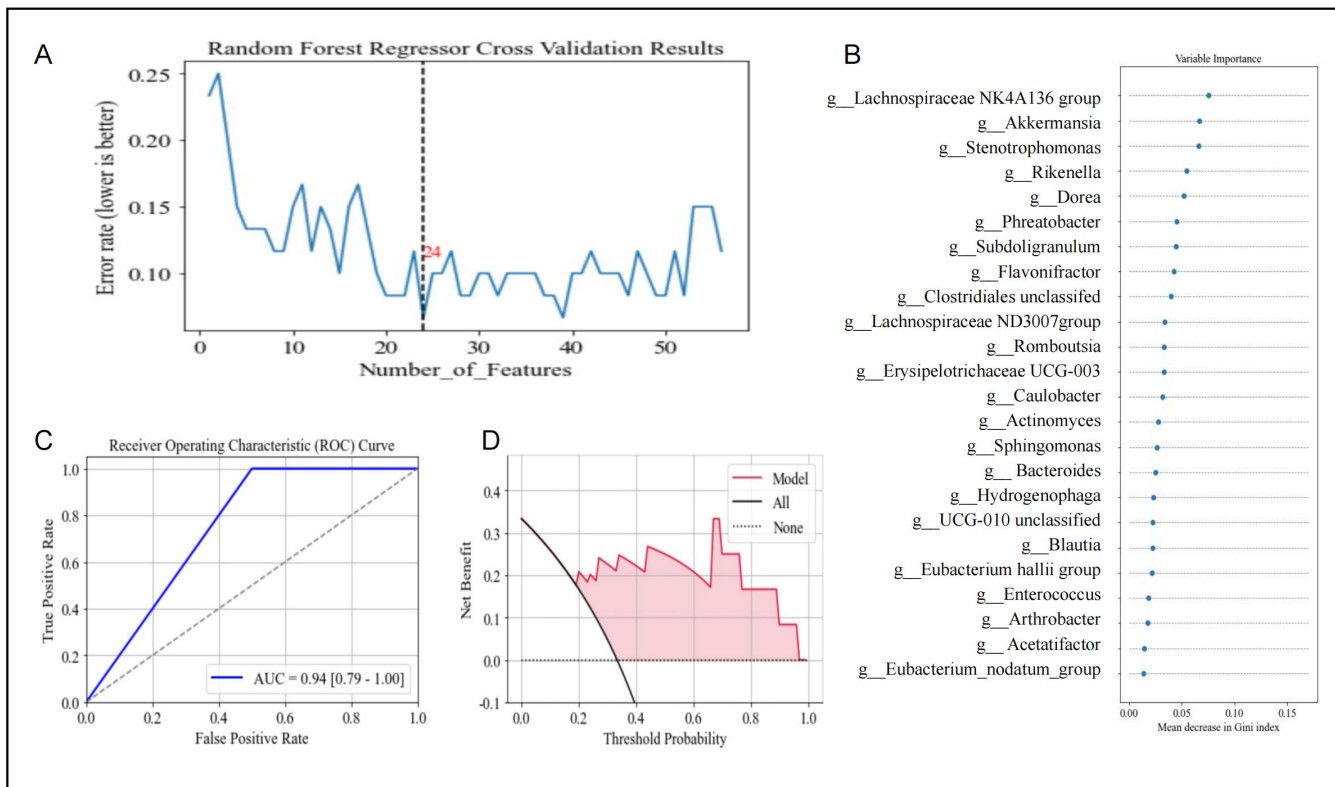

**FIG 4** Random forest model using 24 genera as a classifier can differentiate patients with AVS from HC. (A) Plot showing the relationship between the number of genera and error rates. As the number of genera increased, error rates dropped sharply. The dashed gray line marks the optimal cut-off for biomarker selection, identifying 24 genera as the best predictor count. (B) Variable importance of the genera, assessed using the random forest method in R, with the top 24 most important genera displayed in the plot. (C) ROC curve of the random forest model constructed using the 24 genera. The diagonal line represents an AUC of 0.94 (95% CI = 0.79–1.00). (D) Decision curve analysis (DCA) for the random forest model based on the 24 genera. The y-axis shows the net benefit. The black solid line represents the scenario where all patients received treatment for AVS, whereas the black dotted line shows the net benefit assuming no patients received treatment. The red line consistently lies above the solid and dotted black lines, especially within the threshold probability range of 0.19–0.96, suggesting that the prediction model offers a higher net benefit compared to scenarios without the model, whether patients are treated or not.

SCFAs-producing genera (25–27). Reduced abundance of SCFA-producing bacteria is associated with elevated inflammatory markers in AS by activating PPARα, which downregulates the NF-κB-induced expression of IL-1β and TNF-α (28). Additionally, *Akkermansia*, an anti-inflammatory bacterium (29), was significantly reduced in atherosclerotic Apoe$^{-/-}$ mice, which aligns with our findings. This study also observed an increase in inflammation-related bacteria in the AVS group, including *Megamonas*, *Flavonifractor*, and *Faecalibacterium* (30). Liu et al. (16) discovered that *Megamonas* was enriched in patients with coronary artery disease and played a significant role in distinguishing patients with coronary artery disease from HC. This suggests that *Megamonas* may have a harmful effect in patients with coronary artery disease, which partially supports our findings.

To further explore the role of characteristic microbiota profiles in the occurrence and development of AVS, this study further analyzed the functional information of the gut microbiota. The results showed that, compared to the healthy control group, the abundance of glycolysis/gluconeogenesis-related functional genes was significantly reduced in AVS patients, suggesting that the progression of AVS may be associated with dysregulation of glucose metabolism (31). In addition, functional genes related to transport and catabolism exhibited a similar trend, which is consistent with the findings of Sanchez-Gimenez R (37) and others, who demonstrated the association between gut microbiota-derived metabolites and cardiovascular diseases. These results suggest that the dysregulation of the gut microbiota in AVS may lead to an imbalance in metabolic

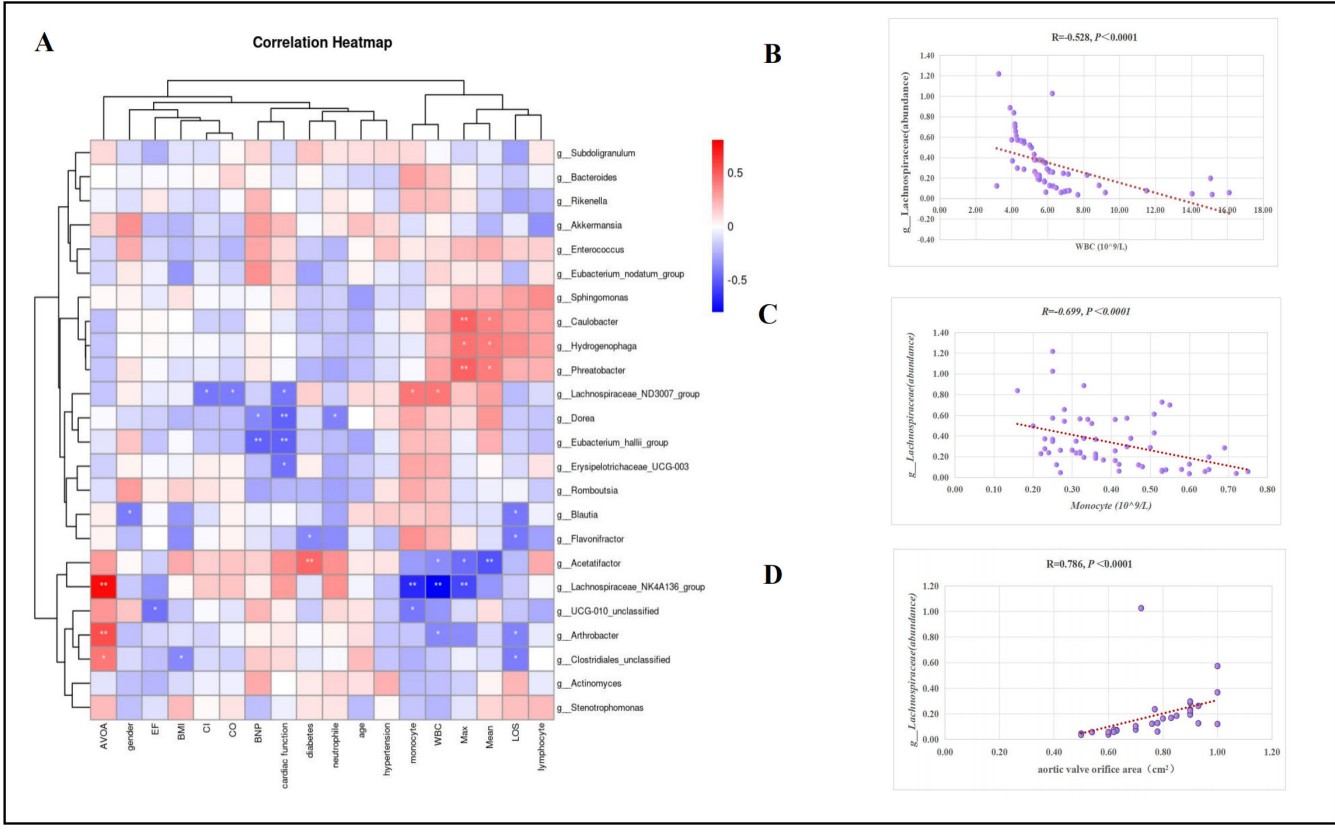

FIG 5 Spearman correlation analysis of different bacteria and clinical characteristics. (A) Heatmap of correlations between bacterial genera and clinical characteristics. (B) Scatter plots showing the correlation between WBC and *Lachnospiraceae*. (C) Scatter plots showing the correlation between monocytes and *Lachnospiraceae*. (D) Scatter plots showing the correlation between AVOA and *Lachnospiraceae*. Color intensity represents the correlation strength: red indicates positive correlations, and blue indicates negative correlations; *$P < 0.05$; **$P < 0.01$, $N = 30:30$.

functions, potentially contributing to the onset and progression of AVS. This will be a key focus for our future research at the animal and cellular levels.

Additionally, Spearman correlation analysis was utilized to further investigate the association between certain GM and clinical features. The study results showed that GM with distinct characteristics in AVS was associated with worsening clinical indicators. Particularly, a negative correlation was observed between *Lachnospiraceae* and inflammatory markers (WBCs and monocytes). As the relative abundance of *Lachnospiraceae* decreased, the levels of inflammatory markers increased. This finding is consistent with a previous study on GM and cardiovascular diseases, which suggests that change in GM (a reduction in the abundance of microorganisms that produce SCFAs) stimulates the activity of inflammatory cytokines (32). Our results also showed that differential microflora (*Lachnospiraceae*) was associated with AVOA. During AVS development, the aortic valve develops calcification, and the CO gradually decreases, with the valve opening area reflecting disease severity (33). As previously reported, various GM and serum metabolites are closely related to the severity of coronary heart disease. Furthermore, GM may influence bile acid (BA) metabolism and contribute to the progression of coronary atherosclerosis (16). Animal studies using fecal microbiota transplantation have provided causal evidence: mice transplanted with pro-inflammatory GM from Caspase1 mice had atherosclerotic plaques that were 29% larger than those in the control group (34). SCFAs have been widely recognized for their anti-inflammatory properties (35). Specifically, they act through G-protein-coupled receptors (e.g., GPR41, GPR43) on immune cells, leading to a reduction in the production of pro-inflammatory cytokines and an increase in anti-inflammatory cytokines. In the context of AVS, chronic inflammation has been shown to contribute to valve calcification and fibrosis

(36). This could slow down the pathological remodeling of the aortic valve, reducing the rate of stenosis. Furthermore, given the known association between lipid accumulation and valve calcification in AVS, alterations in the gut microbiota composition, especially the reduction of SCFA-producing genera, could indirectly promote lipid-driven inflammatory processes that accelerate AVS progression (5, 16). Kocyigit (17) was the first to investigate the relationship between diet, GM-derived metabolites, and calcified aortic stenosis. The study discovered that patients with plasma choline levels (≥14.98 µM) had higher classification scores both in aortic and mitral valves, which were independently associated with peak aortic flow velocity. However, the study by Kocyigit D primarily focused on diet and GM metabolites and did not explore the correlation between GM and AVS. Additionally, the study only matched for sex and age, overlooking the influence of BMI and region on GM.

In summary, we hypothesized that abnormal changes and metabolic imbalance in GM of patients with AVS result in an inflammatory response, playing an essential role in the occurrence of AVS and enhancing its development. However, the exact role of GM in patients with AVS and its underlying molecular mechanisms remains unreported. Therefore, future longitudinal studies in larger cohorts, including the investigation of GM dynamics over time, could validate the clinical relevance of these biomarkers in AVS progression and therapy.

## Limitations and future perspectives

This study has several limitations. First, although participants were matched based on basic dietary patterns and key food intake variables were adjusted for, the absence of detailed dietary records (e.g., food frequency questionnaires) remains a constraint. Future studies should integrate metagenomic sequencing with standardized dietary assessments, such as 24 hour recalls, to better elucidate diet–microbiota–disease interactions. Second, functional predictions based on 16S data using PICRUSt are inherently limited. Therefore, future studies should employ shotgun metagenomics and metabolomics to validate the inferred pathways. Finally, although internal validation was conducted through nested cross-validation, external validation using independent prospective cohorts is necessary to confirm generalizability. Longitudinal designs with serial sampling could help establish causality and clarify temporal dynamics within the gut–heart axis. Further mechanism research is still needed to verify the roles of specific microbiota and their metabolites, as well as their direct impact on cardiac transport proteins through *in vitro* experiments and animal models.

## Conclusions

This study revealed, for the first time, that patients with AVS exhibit a unique GM profile characterized by a reduced abundance of SCFAs-producing bacterial genera. Differential GM demonstrated high predictive potential for AVS and was associated with worsening clinical markers, with 24 genera identified as biomarkers and predictors in clinical applications. This study highlights the previously unknown role of GM in AVS, providing a theoretical foundation for further GM-focused intervention experiments and offering new ideas for the prevention and treatment of AVS.

## ACKNOWLEDGMENTS

The authors appreciate all subjects who participated in this study. We would also like to thank the hospital for supporting the data collection for the study.

The authors acknowledge the following financial support for the research, authorship, and/or publication of this article: the National Natural Science Foundation of China (grant no. 8237050414), the Fujian Provincial Department of Finance Project (2023CZ005), and the Key Laboratory of Cardio-Thoracic Surgery (Fujian Medical University), Fujian Province University Construction Project (No. 2019-67).

F.J. proposed the idea and designed and drafted the manuscript, whereas MLC conducted the data analysis. Y.C.P., J.L.L., and S.L.L. were responsible for data collection, Q.P., B.L., and Y.L.X. handled the collection of fecal samples. Y.J.L. provided scientific insights and critically reviewed the manuscript. L.W.C. supervised the project. All authors have approved the journal to which the article has been submitted and agree to be accountable for all aspects of the work.

The authors appreciate all subjects who participated in this study. We would also like to thank the hospital for supporting the data collection of this study.

## AUTHOR AFFILIATIONS

[1]Department of Nursing, Fujian Medical University Union Hospital, Fuzhou, China
[2]Department of Cardiovascular Surgery, Fujian Medical University Union Hospital, Fuzhou, China
[3]Key Laboratory of Cardio-Thoracic Surgery (Fujian Medical University), Fujian Province University, Fuzhou, China
[4]Department of Physical Examination, Fujian Medical University Union Hospital, Fuzhou, China

## AUTHOR ORCIDs

Liangwan Chen http://orcid.org/0000-0002-4211-3842
Yanjuan Lin http://orcid.org/0000-0001-8953-7549

## FUNDING

| Funder | Grant(s) | Author(s) |
| --- | --- | --- |
| National Natural Science Foundation of China | 8237050414 | Yanjuan Lin |
| Fujian Provincial Department of Science and Technology | 2023CZ005 | Yanjuan Lin |

## AUTHOR CONTRIBUTIONS

Fei Jiang, Methodology, Writing – original draft | Meiling Cai, Data curation, Software, Validation | Yanchun Peng, Project administration, Software | Sailan Li, Project administration, Validation, Visualization | Yuling Xie, Data curation, Methodology | Qiong Pan, Data curation, Project administration | Jianlong Lin, Data curation, Software | Bing Liang, Data curation, Methodology | Liangwan Chen, Supervision, Visualization, Writing – review and editing | Yanjuan Lin, Conceptualization, Funding acquisition, Writing – review and editing

## DATA AVAILABILITY STATEMENT

The original contributions of this study are included in the article, and the data supporting its findings are available from the corresponding author upon reasonable request. The raw 16S rDNA sequencing data generated in this study have been deposited in the NCBI SRA under BioProject accession PRJNA1218297. (https://www.ncbi.nlm.nih.gov/bioproject/PRJNA1218297). These data will be publicly available on December 30, 2025, or the article was published, whichever comes first.

## ETHICS APPROVAL

This clinical study was approved by the Ethics Review Committee of Union Hospital, Fujian Medical University, Fujian, China (Ethics Approval No. 2024KY012). The study protocol adhered to the Declaration of Helsinki. Additionally, we ensured strict compliance with patient confidentiality and data privacy guidelines. All patient data were

anonymized, and informed consent was obtained from all participants before inclusion in the study.

## ADDITIONAL FILES

The following material is available online.

## Supplemental Material

**Supplemental data (Spectrum03215-24-s0001.docx).** Supplementary statistical analyses.

**Supplemental material (Spectrum03215-24-s0002.docx).** Supplemental materials and methods description of 16SrDNA sequencing.

**Table S1 (Spectrum03215-24-s0003.xlsx).** ASV feature sequences of samples.

**Table S2 (Spectrum03215-24-s0004.xlsx).** PICRUSt2 function prediction of samples at KEGG.

## Open Peer Review

**PEER REVIEW HISTORY (review-history.pdf).** An accounting of the reviewer comments and feedback.

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
