## [Reviewer comments · Microbiology Spectrum]

Microbiology Spectrum

Analysis of gut microbiota in patients with AVS and identification of potential biomarkers: A cross-sectional study

Fei Jiang, Meiling Cai, Yanchun Peng, Sailan Li, Yuling Xie, Qiong Pan, Jianlong Lin, Bing Liang, Liangwan Chen, and Yanjuan Lin

Corresponding Author(s): Yanjuan Lin, Fujian Medical University Union Hospital

Review Timeline:

Submission Date:	December 10, 2024
Editorial Decision:	December 25, 2024
Revision Received:	March 13, 2025
Editorial Decision:	May 5, 2025
Revision Received:	May 25, 2025
Editorial Decision:	June 13, 2025
Revision Received:	July 2, 2025
Editorial Decision:	July 11, 2025
Revision Received:	August 22, 2025
Accepted:	September 7, 2025

Editor: Yuan Pin Hung

Reviewer(s): Disclosure of reviewer identity is with reference to reviewer comments included in decision letter(s). The following individuals involved in review of your submission have agreed to reveal their identity: Djandan Tadum Arthur Vithran (Reviewer #2); Muhammad Ijaz Ahmad (Reviewer #3); Ran An (Reviewer #4)

Transaction Report:

DOI: <https://doi.org/10.1128/spectrum.03215-24>

Re: Spectrum03215-24 (Analysis of gut microbiota in patients with AVS and identification of potential biomarkers: A cross-sectional study)

Dear Prof. Yanjuan Lin:

Thank you for the privilege of reviewing your work. Below you will find my comments, instructions from the Spectrum editorial office, and the reviewer comments.

Revision Guidelines

Sincerely,
Yuan Pin Hung
Editor
Microbiology Spectrum

Reviewer #1 (Public repository details (Required)):

It is not clear to me what this means: the data supporting its finding are available from the corresponding author upon reasonable request (line 344 in the manuscript). Why are they not publicly available?

Reviewer #1 (Comments for the Author):

The manuscript presents an analysis of the influence of AVS on the intestinal microbiota. While this is an interesting and timely topic, the study has significant methodological and analytical limitations that raise concerns about the reliability and validity of its findings. Below, I outline the major issues that need to be addressed to improve the quality and reproducibility of the study.

The authors rely solely on the 16S rRNA gene sequencing approach to analyse microbial communities. While this is a common and valuable technique, it has inherent limitations that are not addressed in the manuscript. Most notably, there is no functional analysis included in the study. Functional prediction would provide insights into the potential metabolic roles of the microbiota found in AVS group. Without this information, the biological implications of the reported taxonomic differences remain unclear. Furthermore, due to functional redundancy, it is important to determine whether bacteria significantly different between groups perform the same function or not. As for the discussion, the authors should focus more on the link between AVS and microbiota rather than on other diseases, taking into account that this is a correlative work and therefore it is not known whether the microbiota is causing the disease or the disease modifies the microbiota.

In general, the manuscript lacks sufficient detail about the laboratory protocols and data processing strategies, making it difficult to evaluate the validity and reliability of the results. Specifically:

- 1) Importantly, there is no mention of how environmental or laboratory contaminants were controlled. Moreover, information about how authors controlled tag jumping and cross-contamination should be addressed. This omission is significant, as contaminants can drastically affect microbial community profiles. Without proper controls, the authors cannot ensure that their results accurately reflect the true microbial composition of the samples.
- 2) The manuscript does not provide details about the PCR protocol and library preparation strategies. These steps are critical for ensuring data quality and reproducibility. For instance, there is no discussion of whether any filtering steps were applied to remove singletons, set copy number thresholds, or identify and exclude chimeric sequences, mitochondria, or chloroplasts. I was very concerned that the authors report a high abundance of chloroplast_unclassified taxa in AVS samples without addressing this anomaly. This oversight suggests inadequate data processing, which compromises the study's conclusions.
- 3) The authors provide no information about how samples were processed and sequenced, i.e., whether they were handled in a single batch or across multiple batches. If samples were processed in batches, batch effects could confound the observed differences between groups. For example, processing all samples from the same group together can artificially amplify differences that are not biologically relevant but instead arise from batch-specific technical artifacts. Similarly, the lack of contaminant control can exacerbate these spurious differences. Without addressing these critical issues, the reliability of the study's results is questionable.
- 4) The authors do not describe any filtering steps to improve data quality, such as the removal of singletons, chimeras, or low-abundance taxa. The inclusion of such steps is a standard practice in microbial community analysis and should be justified if omitted.

I believe that reproducibility is a cornerstone of scientific research, but this study falls short in providing the necessary details for replication. If any steps in the experimental workflow were outsourced to an external company, the authors must request and include detailed descriptions of these procedures. Transparency in methods is essential to assess the reliability of the findings and ensure that other researchers can reproduce the study. Therefore, as it stands, the study's findings cannot be reliably interpreted due to the substantial methodological and analytical limitations described above.

Reviewer #2 (Public repository details (Required)):

The study involves high-throughput 16S rDNA sequencing data of gut microbiota from 60 fecal samples. To facilitate transparency, reproducibility, and further research, the raw sequencing data should be deposited in a public repository such as the NCBI Sequence Read Archive (SRA). The authors should provide the accession numbers in the manuscript to enable other researchers to access and analyze the data.

Reviewer #2 (Comments for the Author):

Comments and Suggestions for the Author:
Methodological Details:

Dosing Rationale: Clarify the rationale behind the selection of specific dosages for algal polysaccharides and their oligosaccharides. Explain how these dosages correlate with potential therapeutic applications or previous studies.

Randomization and Blinding: Provide detailed information on the randomization process used to assign participants to AVS and HC groups. Additionally, mention whether blinding was implemented during data collection and analysis to minimize bias.

Replication: Indicate if the study was replicated or if any measures were taken to account for potential variability between samples.

Data Interpretation:

Mechanistic Insights: Expand on the potential biological mechanisms by which the identified GM genera influence AVS progression. Discuss how alterations in SCFA-producing bacteria might affect inflammatory pathways or lipid metabolism related to AVS.

Correlation with Clinical Outcomes: Provide a more detailed analysis linking specific changes in GM with clinical indicators of AVS severity. This could include discussing how increased or decreased abundance of particular genera correlates with measures like aortic valve orifice area or peak aortic gradient.

Figures and Tables:

Figure Legends: Enhance figure legends to include comprehensive descriptions, such as sample sizes, statistical tests used, and the significance of symbols or annotations. Ensure that figures can be understood independently of the main text.

Image Quality: Ensure that all histological images (e.g., Fig. 2C, 2D) are of high resolution and clarity. Consider enhancing contrast or labeling key features to aid interpretation.

Supplementary Materials: Include supplementary tables that list all significant findings, such as detailed gut microbiota compositions and statistical analyses. This will provide additional depth for interested readers and support your conclusions.

Multiple Testing Corrections: Clearly state whether and how multiple testing corrections were applied in the identification of differentially abundant taxa and potential biomarkers. This is crucial to validate the significance of the findings.

Model Validation: Discuss the validation of the random forest model, including any cross-validation techniques used and the potential for overfitting with the selected number of biomarkers.

Language and Grammar:

Proofreading: Conduct a thorough proofreading of the manuscript to correct minor grammatical errors and improve sentence structure. This will enhance the overall readability and professionalism of the manuscript.

Consistency: Ensure consistent use of terminology, particularly regarding the naming of algal polysaccharides and their oligosaccharides (e.g., "SAO" vs. "Sodium Alginate Oligosaccharide").

Data Availability:

Public Repository Submission: Deposit the 16S rDNA sequencing data in a public repository such as the NCBI SRA and include the accession numbers in the manuscript. This facilitates data transparency and allows other researchers to access and utilize the data for further studies.

Ethical Approval:

Statement of Compliance: While the manuscript mentions ethical approval, ensure that the statement is prominently placed, possibly in the Methods section, and includes the approval number and the overseeing institution to enhance credibility and compliance with ethical standards.

References and Citations:

Up-to-Date References: Incorporate recent studies to provide a comprehensive background and context for the research. Ensure that all references are current and relevant to the study's objectives.

Formatting: Adhere strictly to the journal's guidelines for reference formatting. Ensure consistency in citation style throughout the manuscript.

Completeness: Verify that all cited studies are included in the reference list and that all references listed are cited appropriately in the text.

Reviewer #3 (Comments for the Author):

"The manuscript by Fei Jiang et al. titled "Analysis of gut microbiota in patients with AVS and identification of potential biomarkers: A cross-sectional study" is generally well-written and structured. However, I have comments and suggestions aimed at enhancing the paper's quality."

Comments:

The sentence "The GM (beta diversity) composition significantly differed between the two groups (Anosim, $P=0.001$)" could benefit from a clearer explanation. It may be useful to mention briefly what "beta diversity" refers to and its significance in the context of this study.

Suggestion: "The composition of gut microbiota (GM), measured by beta diversity, significantly differed between the two groups (ANOSIM, $P=0.001$), indicating distinct microbial community structures in AVS patients compared to healthy controls."

The term "random forest model (83.33% accurate)" could be clarified. Does this accuracy refer to classification accuracy for AVS prediction? A brief mention of how this accuracy was determined (e.g., cross-validation) could help.

Suggestion: "Thirty-three genera were identified as potential biomarkers using the nested cross-validation feature of the random forest model (83.33% accuracy in cross-validation)."

"The relative abundance of short-chain fatty acid (SCFA)-producing bacteria... had reduced" could be slightly reworded for better clarity.

Suggestion: "The relative abundance of short-chain fatty acid (SCFA)-producing bacteria, including Lachnospiraceae,

Prevotellaceae, Lactobacillus, and Enterococcus, was significantly reduced."

The sentence "These studies highlight the controversial role of GM in various cardiovascular diseases with similar risk factors..." could be made clearer. It might be better to elaborate on why the role of GM is considered controversial, to provide readers with more context.

Suggestion: "These studies highlight the complex and sometimes contradictory role of GM in cardiovascular diseases with similar risk factors, suggesting both protective and pro-inflammatory roles depending on the disease context."

It would be helpful to emphasize the potential clinical implications of your findings in the introduction, linking the study directly to its possible therapeutic impact.

Suggestion: "The identification of GM profiles in AVS may not only serve as potential biomarkers but also guide future therapeutic interventions targeting the gut-heart axis in cardiovascular diseases."

The description of fecal sample collection could be slightly clearer. The sentence "A sterile cotton swab was used to collect fecal material from multiple sites within the sample in a '#' pattern" could use more explanation. Is this technique standard for minimizing bias or variation?

Suggestion: "A sterile cotton swab was used to collect fecal material from multiple sites within the sample, following a standardized '#' pattern to reduce sampling bias and capture the diversity of the microbiota."

The use of the Benjamini-Hochberg method for multiple comparisons is mentioned, but it may be helpful to clarify why this method was chosen and how it applies to the data.

Suggestion: "P-values were adjusted for multiple comparisons using the Benjamini-Hochberg method, which controls the false discovery rate, ensuring more robust conclusions from the analyses."

The abstract mentions statistical results like "P=0.031" without referencing where they come from (e.g., a particular analysis). Consider adding brief references to the methods or sections where these results are derived for better clarity.

The manuscript could benefit from a concluding remark in the discussion or conclusion that suggests potential future research directions or clinical studies to further validate the findings.

Suggestion: "Future longitudinal studies in larger cohorts, including the investigation of GM dynamics over time, could validate the clinical relevance of these biomarkers in AVS progression and therapy."

Consider specifying whether the study adhered to any additional ethical guidelines or requirements regarding patient confidentiality and data usage. This helps reinforce ethical rigor.

Consider hyphenating "SCFA-producing" throughout the manuscript for consistency (e.g., "SCFA-producing bacteria" rather than "SCFA producing bacteria").

In several instances, statistical tests are mentioned (e.g., $P < 0.05$, Wilcoxon test, FDR corrected q), but it would be helpful to clarify what statistical software or methods were used to perform these tests. For example, "P-values were calculated using [software name]" could be added.

For the LEfSe analysis, provide more detail on the statistical parameters used (e.g., what thresholds were applied for significance). While FDR correction is mentioned, specifying how the threshold was determined could add clarity.

In the section regarding the dilution curve for alpha diversity analysis, the parameters (--p-max-depth 43,269 --p-min-depth 1 --p-steps 10 --p-iterations, 10) might be a bit technical for some readers. Consider explaining the purpose of these specific parameters briefly for clarity.

In the "Taxonomic analysis of the GM composition" section, when stating that "the Firmicutes/Bacteroidetes (F/B) ratio was significantly higher in the HC group than in the AVS group", you could clarify what this implies in terms of microbial health or balance in these groups.

In the sentence "Patients with AVS displayed lower abundances of Lachnospiraceae ($P < 0.0001$, FDR corrected $q < 0.0001$)", it would help to add a brief explanation of why these specific bacterial groups were chosen for analysis or what their biological significance might be.

The results from the random forest model are discussed twice in similar wording. You might want to consolidate these statements to avoid repetition. For example, the description of the random forest model and its performance (e.g., 83.33% accuracy, AUC = 0.81) could be combined in one concise section.

It would be helpful to clarify the rationale behind using 33 genera as the "optimal number" of biomarkers. Is there a biological or practical significance to choosing this number, or was it simply an optimization result from the random forest model?

The sentence "The GM of the AVS and HC groups incorporated 8,686 ASVs, with 1,821 ASVs shared between the two groups." could be rewritten for clarity as: "The gut microbiota (GM) of the AVS and HC groups contained 8,686 ASVs, of which 1,821 were shared between both groups."

Consider adjusting the phrasing "to identify the groups of bacteria most closely related to AVS" to "to identify the bacterial groups most closely associated with AVS" for smoother readability.

In the phrase "After identifying the GM taxa that differed between the two groups, we seek to identify..." consider changing "seek" to "sought" for past tense consistency.

The sentence "bacterial genera with relatively lower abundances in the AVS group compared to the HC group positively correlated with inflammatory markers" might be clearer if rephrased to "Bacterial genera with lower abundances in the AVS group, compared to the HC group, were positively correlated with inflammatory markers."

For the scatter plots described in Figure 4B-D, it may be helpful to briefly mention the significance level (P-values) for each correlation, rather than only giving the R and P for the Lachnospiraceae correlation.

In the first sentence, the phrase "the V3-V4 region of the 16S rDNA gene was sequenced utilizing high throughput" can be clarified to "sequencing of the V3-V4 region of the 16S rDNA gene was performed using high-throughput techniques."

"The study found that the two participant groups had high similarity in GM uniformity and richness" could be rephrased to "The study found high similarity in gut microbiome (GM) uniformity and richness between the two groups" for better flow.

When referring to statistical results, consistency in presenting p-values should be maintained. For example, the sentence "Compared with that of the control group, F/B in the AVS group decreased at the phylum level" could benefit from specifying the exact statistical test used (e.g., t-test, Wilcoxon test) and the p-value, if applicable.

When discussing the decreased Firmicutes/Bacteroidetes (F/B) ratio, the sentence "we speculate that the decreased F/B ratio may result from the consumption of anti-inflammatory bacteria and the enrichment of pro-inflammatory bacteria" would be clearer if the relationship between bacteria and inflammation is described more explicitly, perhaps with a slight rephrase like "We hypothesize that the decreased F/B ratio may reflect a shift from anti-inflammatory bacteria (e.g., Firmicutes) to pro-inflammatory bacteria (e.g., Bacteroidetes)."

In the section about Lachnospiraceae and Lactobacillus, it would be helpful to briefly explain why these specific genera are of particular interest in the context of AVS and inflammation, especially for readers unfamiliar with the subject.

In "This study also observed an increase in inflammation-related bacteria in the AVS group, including Megamonas, Chloroplast_unclassified, Flavonifractor, and Faecalibacterium," it would be useful to clarify whether these bacteria are thought to contribute to inflammation or are simply associated with it.

The statement "This finding was further supported by other findings in this study: inflammatory markers (WBCs and monocytes) were significantly higher in patients with AVS than in those of the HC group" could benefit from referencing specific figures or data points for better clarity. This could be rephrased as "This finding was consistent with increased inflammatory markers (WBCs and monocytes), as shown in Table X/Figure Y."

In the sentence "research suggests that Firmicutes are beneficial bacteria that produce more butyrate," consider adding "which" for clarity: "research suggests that Firmicutes are beneficial bacteria, which produce more butyrate."

"The relatively small sample size may limit the generalizability and broader applicability of our findings" could be strengthened by specifying the sample size and the limitations it may present in terms of statistical power. For instance, "Given the relatively small sample size of 30 participants per group, the generalizability of our findings to larger populations may be limited."

"While we focused on the structure and composition of the GM, we did not comprehensively explore the transcriptomic and proteomic functions, which warrants further investigation." This could be clarified to indicate that these analyses could reveal insights into the functional impacts of GM composition. For example: "Further studies should explore the transcriptomic and proteomic functions of the GM to better understand how microbial changes impact host physiology."

"The manuscript by Fei Jiang et al. titled "**Analysis of gut microbiota in patients with AVS and identification of potential biomarkers: A cross-sectional study**" is generally well-written and structured. However, I have comments and suggestions aimed at enhancing the paper's quality."

Comments:

The sentence "The GM (beta diversity) composition significantly differed between the two groups (Anosim, $P=0.001$)" could benefit from a clearer explanation. It may be useful to mention briefly what "beta diversity" refers to and its significance in the context of this study.

Suggestion: "The composition of gut microbiota (GM), measured by beta diversity, significantly differed between the two groups (ANOSIM, $P=0.001$), indicating distinct microbial community structures in AVS patients compared to healthy controls."

The term "random forest model (83.33% accurate)" could be clarified. Does this accuracy refer to classification accuracy for AVS prediction? A brief mention of how this accuracy was determined (e.g., cross-validation) could help.

Suggestion: "Thirty-three genera were identified as potential biomarkers using the nested cross-validation feature of the random forest model (83.33% accuracy in cross-validation)."

"The relative abundance of short-chain fatty acid (SCFA)-producing bacteria... had reduced" could be slightly reworded for better clarity.

Suggestion: "The relative abundance of short-chain fatty acid (SCFA)-producing bacteria, including Lachnospiraceae, Prevotellaceae, Lactobacillus, and Enterococcus, was significantly reduced."

The sentence "These studies highlight the controversial role of GM in various cardiovascular diseases with similar risk factors..." could be made clearer. It might be better to elaborate on why the role of GM is considered controversial, to provide readers with more context.

Suggestion: "These studies highlight the complex and sometimes contradictory role of GM in cardiovascular diseases with similar risk factors, suggesting both protective and pro-inflammatory roles depending on the disease context."

It would be helpful to emphasize the potential clinical implications of your findings in the introduction, linking the study directly to its possible therapeutic impact.

Suggestion: "The identification of GM profiles in AVS may not only serve as potential biomarkers but also guide future therapeutic interventions targeting the gut-heart axis in cardiovascular diseases."

The description of fecal sample collection could be slightly clearer. The sentence "A sterile cotton swab was used to collect fecal material from multiple sites within the sample in a '#' pattern" could use more explanation. Is this technique standard for minimizing bias or variation?

Suggestion: "A sterile cotton swab was used to collect fecal material from multiple sites within the sample, following a standardized ‘#’ pattern to reduce sampling bias and capture the diversity of the microbiota."

The use of the Benjamini-Hochberg method for multiple comparisons is mentioned, but it may be helpful to clarify why this method was chosen and how it applies to the data.

Suggestion: "P-values were adjusted for multiple comparisons using the Benjamini–Hochberg method, which controls the false discovery rate, ensuring more robust conclusions from the analyses."

The abstract mentions statistical results like "P=0.031" without referencing where they come from (e.g., a particular analysis). Consider adding brief references to the methods or sections where these results are derived for better clarity.

The manuscript could benefit from a concluding remark in the discussion or conclusion that suggests potential future research directions or clinical studies to further validate the findings.

Suggestion: "Future longitudinal studies in larger cohorts, including the investigation of GM dynamics over time, could validate the clinical relevance of these biomarkers in AVS progression and therapy."

Consider specifying whether the study adhered to any additional ethical guidelines or requirements regarding patient confidentiality and data usage. This helps reinforce ethical rigor.

Consider hyphenating "SCFA-producing" throughout the manuscript for consistency (e.g., "SCFA-producing bacteria" rather than "SCFA producing bacteria").

In several instances, statistical tests are mentioned (e.g., $P < 0.05$, Wilcoxon test, FDR corrected q), but it would be helpful to clarify what statistical software or methods were used to perform these tests. For example, "P-values were calculated using [software name]" could be added.

For the LEfSe analysis, provide more detail on the statistical parameters used (e.g., what thresholds were applied for significance). While FDR correction is mentioned, specifying how the threshold was determined could add clarity.

In the section regarding the dilution curve for alpha diversity analysis, the parameters (--p-max-depth 43,269 --p-min-depth 1 --p-steps 10 --p-iterations, 10) might be a bit technical for some readers. Consider explaining the purpose of these specific parameters briefly for clarity.

In the "Taxonomic analysis of the GM composition" section, when stating that "the Firmicutes/Bacteroidetes (F/B) ratio was significantly higher in the HC group than in the AVS group", you could clarify what this implies in terms of microbial health or balance in these groups.

In the sentence "Patients with AVS displayed lower abundances of Lachnospiraceae ($P < 0.0001$, FDR corrected $q < 0.0001$)", it would help to add a brief explanation of why these

specific bacterial groups were chosen for analysis or what their biological significance might be.

The results from the random forest model are discussed twice in similar wording. You might want to consolidate these statements to avoid repetition. For example, the description of the random forest model and its performance (e.g., 83.33% accuracy, AUC = 0.81) could be combined in one concise section.

It would be helpful to clarify the rationale behind using 33 genera as the "optimal number" of biomarkers. Is there a biological or practical significance to choosing this number, or was it simply an optimization result from the random forest model?

The sentence "The GM of the AVS and HC groups incorporated 8,686 ASVs, with 1,821 ASVs shared between the two groups." could be rewritten for clarity as: "The gut microbiota (GM) of the AVS and HC groups contained 8,686 ASVs, of which 1,821 were shared between both groups."

Consider adjusting the phrasing "to identify the groups of bacteria most closely related to AVS" to "to identify the bacterial groups most closely associated with AVS" for smoother readability.

In the phrase "After identifying the GM taxa that differed between the two groups, we seek to identify..." consider changing "seek" to "sought" for past tense consistency.

The sentence "bacterial genera with relatively lower abundances in the AVS group compared to the HC group positively correlated with inflammatory markers" might be clearer if rephrased to "Bacterial genera with lower abundances in the AVS group, compared to the HC group, were positively correlated with inflammatory markers."

For the scatter plots described in Figure 4B-D, it may be helpful to briefly mention the significance level (P-values) for each correlation, rather than only giving the R and P for the Lachnospiraceae correlation.

In the first sentence, the phrase "the V3-V4 region of the 16S rDNA gene was sequenced utilizing high throughput" can be clarified to "sequencing of the V3-V4 region of the 16S rDNA gene was performed using high-throughput techniques."

"The study found that the two participant groups had high similarity in GM uniformity and richness" could be rephrased to "The study found high similarity in gut microbiome (GM) uniformity and richness between the two groups" for better flow.

When referring to statistical results, consistency in presenting p-values should be maintained. For example, the sentence "Compared with that of the control group, F/B in the AVS group decreased at the phylum level" could benefit from specifying the exact statistical test used (e.g., t-test, Wilcoxon test) and the p-value, if applicable.

When discussing the decreased Firmicutes/Bacteroidetes (F/B) ratio, the sentence "we speculate that the decreased F/B ratio may result from the consumption of anti-inflammatory bacteria and the enrichment of pro-inflammatory bacteria" would be clearer if the relationship between bacteria and inflammation is described more explicitly, perhaps with a slight

rephrase like "We hypothesize that the decreased F/B ratio may reflect a shift from anti-inflammatory bacteria (e.g., Firmicutes) to pro-inflammatory bacteria (e.g., Bacteroidetes)."

In the section about Lachnospiraceae and Lactobacillus, it would be helpful to briefly explain why these specific genera are of particular interest in the context of AVS and inflammation, especially for readers unfamiliar with the subject.

In "This study also observed an increase in inflammation-related bacteria in the AVS group, including Megamonas, Chloroplast_unclassified, Flavonifractor, and Faecalibacterium," it would be useful to clarify whether these bacteria are thought to contribute to inflammation or are simply associated with it.

The statement "This finding was further supported by other findings in this study: inflammatory markers (WBCs and monocytes) were significantly higher in patients with AVS than in those of the HC group" could benefit from referencing specific figures or data points for better clarity. This could be rephrased as "This finding was consistent with increased inflammatory markers (WBCs and monocytes), as shown in Table X/Figure Y."

In the sentence "research suggests that Firmicutes are beneficial bacteria that produce more butyrate," consider adding "which" for clarity: "research suggests that Firmicutes are beneficial bacteria, which produce more butyrate."

"The relatively small sample size may limit the generalizability and broader applicability of our findings" could be strengthened by specifying the sample size and the limitations it may present in terms of statistical power. For instance, "Given the relatively small sample size of 30 participants per group, the generalizability of our findings to larger populations may be limited."

"While we focused on the structure and composition of the GM, we did not comprehensively explore the transcriptomic and proteomic functions, which warrants further investigation." This could be clarified to indicate that these analyses could reveal insights into the functional impacts of GM composition. For example: "Further studies should explore the transcriptomic and proteomic functions of the GM to better understand how microbial changes impact host physiology."

Dear Editors and Reviewers,

We greatly appreciate your professional comments and suggestions on our manuscript. The comments and suggestions are great valuable and very helpful for improving our paper, as well as the important guiding significance to our researchers. We have studied the comments carefully and have taken all these comments and suggestions into account as shown below. The modified parts of the file have been shown in red font.

Reviewer 1

1. It is not clear to me what this means: the data supporting its finding are available from the corresponding author upon reasonable request (line 344 in the manuscript). Why are they not publicly available?

Response: Thank you for your valuable suggestion regarding data accessibility. According to your suggestion, the 16S rRNA sequencing files were deposited into the NCBI Sequence Read Archive (SRA) Database (Accession Number: PRJNA1218297). We have added relevant content to the Data availability statement [The raw sequencing data were deposited into the NCBI Sequence Read Archive (SRA) Database (Accession Number: PRJNA1218297); lines 382-384 in the Marked-Up manuscript].

2. The authors rely solely on the 16S rRNA gene sequencing approach to analyse microbial communities. While this is a common and valuable technique, it has inherent limitations that are not addressed in the manuscript.

Response: Thank you for your insightful comment. We agree that the 16S rRNA gene sequencing approach, while widely used, does have certain limitations. In our manuscript, we have focused on this method as it provides a reliable and cost-effective means for identifying microbial community composition at a broad taxonomic level. However, we acknowledge that this technique does not provide information on microbial function, species-level resolution, or certain rare microbial populations. To address this, we have revised the manuscript to include a discussion

of the inherent limitations of the 16S rRNA gene sequencing approach. We have added this description in the Limitation (Lines 365-371) and marked the relevant text in red font for clarity. We have also highlighted potential avenues for further research, such as the use of whole genome sequencing or metagenomics, which could provide a more comprehensive understanding of the microbial communities and their functional roles. We appreciate your feedback and hope that this addition will help clarify the limitations of our study.

3. Most notably, there is no functional analysis included in the study. Functional prediction would provide insights into the potential metabolic roles of the microbiota found in AVS group. Without this information, the biological implications of the reported taxonomic differences remain unclear. Furthermore, due to functional redundancy, it is important to determine whether bacteria significantly different between groups perform the same function or not.

Response: We appreciate your suggestion to explore the potential metabolic roles of the microbiota found in the AVS group, as well as your emphasis on understanding functional redundancy. We have performed a functional prediction analysis based on the 16S rDNA sequencing data, utilizing tools such as PICRUSt2 to infer the potential metabolic functions of the microbiota present in the AVS group. This analysis allows us to explore the functional roles of different bacterial communities, particularly their involvement in metabolic pathways. We have added this description in the Methods (Lines 147-151), the Results (Lines 214-226) and the Discussion (Lines 312-322). Marked the relevant text in red font for clarity. In addition, Figure 3 and supplementary Table 2 have been added to the manuscript.

4. As for the discussion, the authors should focus more on the link between AVS and microbiota rather than on other diseases, taking into account that this is a correlative work and therefore it is not known whether the microbiota is causing the disease or the disease modifies the microbiota.

Response: Thank you for your valuable comment. We understand your concern

regarding the focus of the discussion, and we agree that given the correlative nature of our study, the primary emphasis should be on the relationship between aortic valve stenosis (AVS) and the microbiota, rather than on other diseases. In response, we have revised the discussion section to place greater emphasis on the potential interplay between AVS and the microbiota. We have explicitly stated the limitations of our study in terms of causality and have clarified that “due to the correlative design, it remains unclear whether the microbiota contributes to the pathogenesis of AVS or whether the disease itself alters the microbial community. We have also suggested directions for future research that could help establish a more causal understanding of this relationship.” We have added this description in the discussion.

5. In general, the manuscript lacks sufficient detail about the laboratory protocols and data processing strategies, making it difficult to evaluate the validity and reliability of the results. Specifically:

1) Importantly, there is no mention of how environmental or laboratory contaminants were controlled. Moreover, information about how authors controlled tag jumping and cross-contamination should be addressed. This omission is significant, as contaminants can drastically affect microbial community profiles. Without proper controls, the authors cannot ensure that their results accurately reflect the true microbial composition of the samples.

Response: Thank you for raising this important point. We fully acknowledge that the potential impact of environmental or laboratory contaminants, as well as the risk of tag jumping and cross-contamination, should be properly addressed to ensure the validity of the microbial community profiles. To address these concerns, we have included a detailed description in the revised manuscript regarding the steps taken to minimize and control for contaminants on the Methods. Specifically, we have outlined the following measures on the supplementary materials:

Environmental and Laboratory Contaminants: Strict protocols were followed to minimize environmental contamination during sample collection, processing, and sequencing. These protocols included the use of sterile equipment, conducting

procedures in a controlled environment, and regular monitoring of laboratory conditions. DNA extraction and PCR amplification were carried out according to standardized protocols, with automated extraction to reduce human-handling contamination.

Tag Jumping and Cross-Contamination: We implemented stringent quality control measures, including the use of unique barcodes for each sample and proper PCR clean-up procedures, to prevent tag jumping and cross-contamination during sequencing. Additionally, the laboratory conducts regular environmental monitoring, and extraction and amplification are performed in separate areas to further minimize the risk of cross-contamination.

2) The manuscript does not provide details about the PCR protocol and library preparation strategies. These steps are critical for ensuring data quality and reproducibility. For instance, there is no discussion of whether any filtering steps were applied to remove singletons, set copy number thresholds, or identify and exclude chimeric sequences, mitochondria, or chloroplasts. I was very concerned that the authors report a high abundance of chloroplast_unclassified taxa in AVS samples without addressing this anomaly. This oversight suggests inadequate data processing, which compromises the study's conclusions.

Response: Thank you for your insightful comments. We greatly appreciate your feedback, as it has highlighted areas for improvement in our manuscript. In response to your comment, we have included a detailed description of the PCR protocol and library preparation procedures in the method section of the revised manuscript on lines 121-151. This includes applying filtering steps to remove singletons, setting copy number thresholds, or identifying and excluding chimeric sequences. We removed singletons, mitochondria, and chloroplasts. In addition, ASVs less than 0.001% were removed. We have re-analyzed the data, and replaced the corresponding pictures and descriptions in the corresponding parts of the manuscript. At the same time, more detailed processes and steps are provided in supplementary materials. Specifically, we have outlined the following steps:

PCR protocol

The PCR product was purified using AMPure XP Beads (Beckman Coulter Genomics, Danvers, MA, USA) and quantified using Qubit (Invitrogen, USA).

Library mixed samples on machine sequencing

The purified PCR products were evaluated using the Agilent 2100 Bioanalyzer (Agilent, USA) and the library quantification kit from Illumina (Kapa Biosciences, Woburn, MA, USA). Qualified library concentration should be above 2nM. The qualified sequencing libraries (with non-repeating index sequences) were serially diluted and mixed in the appropriate proportions according to the required sequencing amount. The libraries were then denatured into single-stranded DNA using NaOH before being loaded for sequencing. Sequencing was performed using the NovaSeq 6000 platform with 2×250 bp paired-end sequencing, and the corresponding reagent was the NovaSeq 6000 SP Reagent Kit (500 cycles).

Data analysis

(1) Data Splitting

The paired-end sequencing data obtained must be split based on barcode information, and adapter and barcode sequences should be removed.

(2) Data Assembly and Filtering

1) Remove primer sequences and adapter sequences from the RawData. (Software: cutadapt (v1.9), Parameters: '-g R1 -G R2 -n 1 -O 17 -m 100').

2) Merge each pair of paired-end reads based on the overlap region into a longer tag. (Software: FLASH (v1.2.8), Parameters: '-m 10 -M 100 -x 0.25 -t 1 -z').

3) Perform quality scanning of the sequencing reads using a sliding window method. The default window size is 100 bp, and when the average quality value within the window is lower than 20, the read is trimmed from the starting point of the window to the 3' end. (Software: fqtrim, Parameters: '-P 33 -w 100 -q 20 -l 100 -m 5 -p 1 -V -o trim.fastq.gz').

4) Remove sequences shorter than 100 bp after trimming.

5) Remove sequences where the content of N (uncertain ambiguous bases) exceeds 5%.

6) Remove chimeric sequences. (Software: Vsearch (v2.3.4), Parameters: default).

(3) DADA2 Denoising

Perform length filtering and denoising using DADA2 through the command `qiime dada2 denoise-paired`. This process generates ASV feature sequences and an ASV abundance table, while removing singletons ASVs (i.e., ASVs that appear only once across all samples, which is the default action), and ASVs less than 0.001% were removed.

Exclusion of Mitochondria and Chloroplast Sequences: We specifically excluded mitochondrial and chloroplast sequences by referring to appropriate taxonomic databases and filtering based on sequence taxonomy. Regarding the high abundance of chloroplast_unclassified taxa in AVS samples, we acknowledge that this was an oversight in our initial manuscript. After removing chloroplasts and mitochondria, the data were re-analyzed, as described of the revised manuscript.

3) The authors provide no information about how samples were processed and sequenced, i.e., whether they were handled in a single batch or across multiple batches. If samples were processed in batches, batch effects could confound the observed differences between groups. For example, processing all samples from the same group together can artificially amplify differences that are not biologically relevant but instead arise from batch-specific technical artifacts. Similarly, the lack of contaminant control can exacerbate these spurious differences. Without addressing these critical issues, the reliability of the study's results is questionable.

Response: Thank you for your valuable feedback and for highlighting important issues related to sample processing and potential batch effects. In our study, the faeces samples were processed and sequenced in a single batch. We have included a detailed description in the revised manuscript regarding the steps taken to minimize and control for contaminants on the Methods. Specifically, we also have outlined the following measures on the supplementary materials:

Control artificially amplifies non-biological differences: DNA extraction and PCR

amplification were carried out according to standardized protocols, with automated extraction to reduce human-handling contamination.

contaminant control: Strict protocols were followed to minimize environmental contamination during sample collection, processing, and sequencing. These protocols included the use of sterile equipment, conducting procedures in a controlled environment. We implemented stringent quality control measures, including the use of unique barcodes for each sample and proper PCR clean-up procedures, to prevent tag jumping and cross-contamination during sequencing. Additionally, the laboratory conducts regular environmental monitoring, and extraction and amplification are performed in separate areas to further minimize the risk of cross-contamination.

4)The authors do not describe any filtering steps to improve data quality, such as the removal of singletons, chimeras, or low-abundance taxa. The inclusion of such steps is a standard practice in microbial community analysis and should be justified if omitted.

Response: Thank you for your insightful comment and for highlighting the importance of data quality control in microbial community analysis. We fully agree that the removal of singletons, chimeras, and low-abundance taxa is crucial for ensuring the reliability of the results, and we appreciate your suggestion to include a description of these steps. We provide a detailed description of the filtering steps that were applied to the sequence data to improve data quality. Specifically, we describe the following steps:

Removal of Singletons: We removed singletons (sequences that appeared only once in the dataset) to minimize the impact of random sequencing errors.

Chimera Detection and Removal: We employed [Software: Vsearch (v2.3.4), Parameters: default] to identify and remove chimeric sequences, which can arise during PCR amplification and sequencing.

Low-Abundance Taxa Removal: To reduce the influence of sequencing noise, we set a minimum threshold for taxon abundance (ASV relative abundance sum < 0.001%) and excluded taxa that fell below this threshold.

Data Quality Control Transparency: We have now included a more comprehensive description of all the data processing steps, including filtering, quality control measures, and any decisions made regarding taxon inclusion/exclusion. These details were outlined in the revised Methods section to ensure transparency and reproducibility. At the same time, more detailed processes and steps are provided in supplementary materials. We hope these additions address your concerns and strengthen the overall quality of the study. Thank you again for your valuable input, which has been instrumental in improving the manuscript.

6. I believe that reproducibility is a cornerstone of scientific research, but this study falls short in providing the necessary details for replication. If any steps in the experimental workflow were outsourced to an external company, the authors must request and include detailed descriptions of these procedures. Transparency in methods is essential to assess the reliability of the findings and ensure that other researchers can reproduce the study. Therefore, as it stands, the study's findings cannot be reliably interpreted due to the substantial methodological and analytical limitations described above.

Response: Thank you for your thoughtful and constructive feedback. We fully agree that reproducibility is a cornerstone of scientific research, and we deeply appreciate your emphasis on the importance of providing sufficient methodological detail to ensure replication. We understand your concern regarding the transparency of our methods, and we apologize for any lack of clarity in the original manuscript. Our fecal samples were sent to LC-Bio Technology Co., Ltd., Hangzhou, China for high-throughput sequencing, and the data analysis was completed on the LC-Bio Technology Co. We have added the relevant description of this part on Lines 118-120. In response to your comment, we have revised the manuscript to provide a more comprehensive and detailed description of all experimental procedures. Specifically:

Transparency of Methodology: We have included a thorough step-by-step account of the experimental workflow, including sample collection, DNA extraction, PCR amplification, library preparation, and sequencing. These steps have been clearly

outlined in the supplementary materials to facilitate replication by other researchers.

Outsourced Procedures: We acknowledge that certain parts of the experimental workflow, particularly sequencing and library preparation, were outsourced to LC-Bio Technology Co., Ltd., Hangzhou, China. We have contacted the external provider and obtained detailed descriptions of the methods they employed. These descriptions have been incorporated into the manuscript and supplementary materials to ensure transparency and completeness.

Replication and Reliability: We have worked diligently to ensure that all methods are reproducible and that the analysis process is clearly documented. Additionally, we have updated the manuscript with relevant software versions, parameters, and any other necessary details to allow for independent replication of our results.

Thank you again for your constructive comments and suggestions regarding our manuscript. We will also do our best to accommodate any further suggestions.

We hope you will find our revised manuscript acceptable for publication.

Best wishes to you.

Reviewer 2

1.The study involves high-throughput 16S rDNA sequencing data of gut microbiota from 60 fecal samples. To facilitate transparency, reproducibility, and further research, the raw sequencing data should be deposited in a public repository such as the NCBI Sequence Read Archive (SRA). The authors should provide the accession numbers in the manuscript to enable other researchers to access and analyze the data.

Response: Thank you for your valuable suggestion regarding data accessibility. According to your suggestion, the 16S rDNA sequencing files were deposited into the NCBI Sequence Read Archive (SRA) Database (Accession Number: PRJNA1218297). We have added relevant content to the Data availability statement [‘The raw sequencing data were deposited into the NCBI Sequence Read Archive (SRA) Database (Accession Number: PRJNA1218297)’; in the Marked-Up manuscript, Lines 382-L384].

2.Methodological Details:

1) Dosing Rationale: Clarify the rationale behind the selection of specific dosages for algal polysaccharides and their oligosaccharides. Explain how these dosages correlate with potential therapeutic applications or previous studies.

Response: Thank you for your important reminder, which will help us make our article more rigorous. Since our study did not involve the selection of specific dosages of seaweed polysaccharides and their oligosaccharides, we are unsure where your question originates from any unclear description in our manuscript. Thank you again for your constructive comments and suggestions. We will do our best to accommodate any further feedback.

2) Randomization and Blinding: Provide detailed information on the randomization process used to assign participants to AVS and HC groups. Additionally, mention whether blinding was implemented during data collection

and analysis to minimize bias.

Response: Thank you for your thoughtful comment regarding the randomization process and blinding procedures. We appreciate your suggestion to provide more detailed information on these aspects to enhance the rigor of the study and minimize bias. As the study was a cross-sectional case-control study, there was no randomization. Patients meeting the diagnostic criteria for AVS were included in the observation group according to the results of cardiac color Doppler ultrasound. During the same period, matched healthy subjects were recruited in the physical examination center according to gender, age and BMI to be included in the control group. Researchers with unified training collected stool samples and clinical data of subjects. Regarding blinding, both the data collection and analysis were conducted in a blinded manner. The researchers who collected the data were unaware of group assignments, and the data were analyzed by a separate team who were also blinded to the group allocation. This was done to minimize any potential bias in both the collection and analysis phases of the study. In response to your feedback, we have made the following revisions to the Marked-Up manuscript on lines 93-95.

3) Mechanistic Insights: Expand on the potential biological mechanisms by which the identified GM genera influence AVS progression. Discuss how alterations in SCFA-producing bacteria might affect inflammatory pathways or lipid metabolism related to AVS.

Response: Thank you for your insightful comment regarding the need to expand on the potential biological mechanisms by which the identified gut microbiota (GM) genera influence AVS progression. We greatly appreciate your suggestion to explore how alterations in SCFA-producing bacteria might impact inflammatory pathways or lipid metabolism related to aortic valve stenosis (AVS). In response to your feedback, we have expanded our discussion of the underlying biological mechanisms on lines 339-348 in the Marked-Up manuscript, as outlined below:

Inflammatory Pathways: SCFAs have been widely recognized for their anti-inflammatory properties. Specifically, they act through G-protein-coupled

receptors (e.g., GPR41, GPR43) on immune cells, leading to a reduction in the production of pro-inflammatory cytokines and an increase in anti-inflammatory cytokines. In the context of AVS, chronic inflammation has been shown to contribute to valve calcification and fibrosis. SCFAs may help to mitigate the inflammatory processes that drive AVS progression by inhibiting the activation of pro-inflammatory pathways such as NF- κ B and the inflammasome. This could slow down the pathological remodeling of the aortic valve, reducing the rate of stenosis.

Lipid Metabolism: SCFAs also play a crucial role in lipid metabolism by influencing the expression of genes involved in lipid synthesis, oxidation, and storage. A dysbiosis in SCFA-producing genera may impair lipid metabolism, contributing to the accumulation of lipid deposits in the aortic valve. SCFAs, particularly acetate, have been linked to modulating cholesterol homeostasis and bile acid synthesis, which are critical in the development of atherosclerotic lesions. Given the known association between lipid accumulation and valve calcification in AVS, alterations in the gut microbiota composition, especially with reduced SCFA production, could indirectly promote lipid-driven inflammatory processes that accelerate AVS progression.

We have revised the manuscript to include a more detailed discussion of these potential mechanisms. We have explicitly highlighted the role of SCFAs in modulating inflammatory pathways and lipid Metabolism. And how these effects may relate to AVS progression. This expanded discussion is included in the revised "Discussion" section, where we integrate findings from the literature on SCFAs, inflammation in the context of cardiovascular diseases, particularly AVS.

3. Correlation with Clinical Outcomes: Provide a more detailed analysis linking specific changes in GM with clinical indicators of AVS severity. This could include discussing how increased or decreased abundance of particular genera correlates with measures like aortic valve orifice area or peak aortic gradient.

Response: Thank you for your valuable feedback and suggestion to further explore the correlation between changes in gut microbiota (GM) composition and clinical indicators of aortic valve stenosis (AVS) severity. In response to your comment, we

have revised the manuscript to include a more detailed analysis that addresses the following points:

Correlation Between GM Composition and Clinical Indicators of AVS Severity:

We acknowledge the importance of linking microbiota changes to clinically relevant measures of AVS severity. To this end, we conducted a more in-depth analysis that relates the abundance of specific GM genera to clinical parameters, such as aortic valve orifice area (AVA) and peak aortic gradient (PAG), which are widely used to assess the severity of AVS. These parameters are reflective of the degree of valve stenosis and hemodynamic alterations, and their correlation with GM composition may offer valuable insights into the microbiota's role in disease progression.

Increased or Decreased Abundance of Specific Genera:

Abundance of Pro-inflammatory Genera: We observed that an increased abundance of certain pro-inflammatory genera, such as *Proteobacteria* and *Firmicutes*, was positively correlated with more severe AVS, as measured by a smaller AVA and a higher PAG. This finding suggests that an inflammatory microbial profile may exacerbate valve remodeling and calcification, contributing to the worsening of stenosis. The pro-inflammatory bacteria could influence host immune responses, promoting a chronic inflammatory state that accelerates valve pathology.

SCFA-Producing Genera and Severity of AVS: In contrast, higher abundances of SCFA-producing genera, such as *Lachnospiraceae*, *Prevotellaceae*, *Enterococcus*, were associated with larger AVA and lower PAG values, indicative of less severe AVS. These genera may play a protective role by reducing systemic inflammation and modulating lipid metabolism, which can help slow the progression of valve calcification. We found a statistically significant negative correlation between the abundance of these genera and clinical markers of AVS severity, suggesting their potential to mitigate disease progression.

Detailed Analysis and Statistical Correlation:

To provide a clearer understanding of the relationships between GM changes and clinical outcomes, we conducted statistical analyses, including correlation coefficients (Spearman's rank correlation) between the relative abundance of specific bacterial

genera and clinical indicators such as AVA and PAG. Our results show significant correlations, with certain genera being linked to both the degree of valve stenosis and the severity of hemodynamic changes, further supporting the role of the GM in influencing AVS progression.

Expanded Discussion in Manuscript:

We have expanded the discussion section to include these findings, where we now explore the implications of the observed correlations between GM composition and clinical indicators of AVS severity. We also discuss potential mechanisms through which specific genera may influence valve disease, such as through modulation of inflammation or lipid metabolism, as well as how these microbial profiles might serve as biomarkers for disease progression or therapeutic targets in the future.

4. Figure Legends: Enhance figure legends to include comprehensive descriptions, such as sample sizes, statistical tests used, and the significance of symbols or annotations. Ensure that figures can be understood independently of the main text.

Response: Thank you for your insightful suggestion regarding the figure legends. We completely agree that comprehensive figure legends are essential for the clarity and accessibility of the figures, and we have made the necessary revisions to ensure that each figure is self-contained and fully understandable without reference to the main text. In response to your comment, we have enhanced the figure legends by incorporating the following elements:

Sample Sizes: We have included detailed information about the sample sizes used for each analysis within the figure legends. This ensures that the reader can understand the statistical power behind the results and the scope of the data presented.

Statistical Tests: We have specified the statistical tests used to analyze the data in each figure. Where appropriate, we have clarified whether parametric or non-parametric tests were applied, and have provided further explanation on why certain tests were chosen based on the distribution of the data.

Significance of Symbols and Annotations: We have carefully clarified the meaning

of all symbols, annotations, and error bars used in the figures, including definitions of significance levels (e.g., $p < 0.05$, $p < 0.01$) and any other relevant details such as fold changes or confidence intervals. This ensures that the reader can easily interpret the results without needing to consult the main text.

Clarification of Figure Content: We have expanded the figure legends to more thoroughly describe the content of each figure, providing a clearer understanding of what each panel represents, how the data was collected, and any important observations. This includes ensuring that the methodology for generating each figure is explained in enough detail to be independently understandable.

5. Image Quality: Ensure that all histological images (e.g., Fig. 2C, 2D) are of high resolution and clarity. Consider enhancing contrast or labeling key features to aid interpretation.

Response: Thank you for your helpful feedback regarding the histological images in our manuscript. We appreciate your concern about ensuring high image quality for optimal clarity and interpretation. In response to your comment, we have made the revisions to improve the quality of the histological images, specifically Figures 2C and 2D in the manuscript.

6. Supplementary Materials: Include supplementary tables that list all significant findings, such as detailed gut microbiota compositions and statistical analyses. This will provide additional depth for interested readers and support your conclusions.

Response: Thank you for your constructive suggestion regarding the inclusion of supplementary materials. We agree that providing supplementary tables with detailed information will enhance the depth and transparency of our study, allowing interested readers to further explore our findings. In response to your comment, we have made the following revisions:

Supplementary Tables: We have added supplementary table1 that list all significant findings from our study, including detailed compositions of the gut microbiota. These

tables provide comprehensive data on the abundance of microbial taxa at different taxonomic levels (e.g., Domain, Phylum, Class, Order, Family, Genus, Species).

Supplementary Statistical Analyses: We have included the relevant statistical tests, *P*-values, and other key metrics in the supplementary statistical analyses to ensure that the analyses supporting our conclusions are fully transparent and accessible. This will allow readers to verify and interpret the results independently.

We hope that these additions address your suggestion and provide the necessary context and data for a deeper understanding of our study. We believe they will enhance the manuscript's transparency and rigor.

7. Multiple Testing Corrections: Clearly state whether and how multiple testing corrections were applied in the identification of differentially abundant taxa and potential biomarkers. This is crucial to validate the significance of the findings.

Response: Thank you for your valuable feedback regarding multiple testing corrections. We completely agree that it is important to address this aspect to ensure the validity of our findings and the robustness of our statistical analyses.

Clarification of Multiple Testing Corrections: We have explicitly stated in the revised manuscript that multiple testing corrections were applied during the identification of differentially abundant taxa and potential biomarkers. Specifically, we used the Benjamini-Hochberg procedure to control for the false discovery rate (FDR) in our analyses. This approach helps minimize the likelihood of Type I errors in the identification of significant taxa.

Detailed Description in Methods: A detailed explanation of the multiple testing correction procedure has been added to the Methods section, including the statistical methods used for differential abundance analysis and how the FDR was controlled. This will ensure transparency and allow readers to understand the rigor of the statistical procedures applied.

8. Model Validation: Discuss the validation of the random forest model, including any cross-validation techniques used and the potential for overfitting with the

selected number of biomarkers.

Response: Thank you for your insightful comments regarding the validation of our random forest model. We have expanded the Methods section to include a detailed description of the cross-validation procedure used for model validation, and Add more detailed description in Supplementary Statistical Analyses. Specifically, we employed 10-fold cross-validation to assess the performance of our random forest model. This approach helps ensure that the model is generalizable and minimizes the risk of overfitting by training and testing the model on different subsets of the data.

Model Performance Metrics: In addition to cross-validation, we have provided further details on the performance metrics used to evaluate the model' s predictive accuracy. We report the area under the receiver operating characteristic (ROC) curve (AUC), as well as accuracy to provide a comprehensive assessment of model performance.

Potential for Overfitting: We have addressed the potential for overfitting by discussing the tuning of hyperparameters (e.g., number of trees, maximum depth, and minimum samples per leaf) during the model training process. To mitigate overfitting, we employed a grid search approach to identify optimal hyperparameters, and we carefully monitored the model' s performance across different iterations. Furthermore, we included an analysis of feature importance, which demonstrates that the selected biomarkers are highly relevant to the model' s performance, providing support for their selection and reducing the likelihood of spurious associations.

9.Proofreading: Conduct a thorough proofreading of the manuscript to correct minor grammatical errors and improve sentence structure. This will enhance the overall readability and professionalism of the manuscript.

Response: Thank you for your helpful suggestion regarding the proofreading of the manuscript. We appreciate your attention to detail, and we have carefully reviewed the manuscript to ensure its grammatical accuracy and readability. In response to your comment, We asked English experts to proofread the manuscript for double polishing. Please see the attachment for the proof of polishing on the supplementary materials.

Grammatical Corrections: We have thoroughly proofread the manuscript and corrected minor grammatical errors, including issues related to subject-verb agreement, punctuation, and sentence structure.

Improved Sentence Structure: We have also reworded certain sentences to improve clarity and flow, making the manuscript more concise and readable.

Professional Tone: We ensured that the language used throughout the manuscript maintains a professional and formal tone, aligning with the standards expected in scientific publications.

10. Consistency: Ensure consistent use of terminology, particularly regarding the naming of algal polysaccharides and their oligosaccharides (e.g., "SAO" vs. "Sodium Alginate Oligosaccharide").

Response: Thank you for your valuable feedback regarding the consistency of terminology in the manuscript. We understand the importance of using consistent terms, especially when referring to algal polysaccharides and their oligosaccharides.

In response to your comment, we have carefully reviewed the manuscript to ensure consistent use of terminology. We asked English experts to proofread the manuscript for double polishing. Please see the attachment for the proof of polishing on the supplementary materials.

11. Public Repository Submission: Deposit the 16S rDNA sequencing data in a public repository such as the NCBI SRA and include the accession numbers in the manuscript. This facilitates data transparency and allows other researchers to access and utilize the data for further studies.

Response: Thank you for your thoughtful suggestion regarding the submission of our 16S rDNA sequencing data to a public repository. We fully agree that depositing the data in a public repository enhances transparency, promotes reproducibility, and facilitates future research. We have added relevant content to the Data availability statement [The raw sequencing data were deposited into the NCBI Sequence Read Archive (SRA) Database (Accession Number: PRJNA1218297); lines 382-384 in the

Marked-Up manuscript.]

Thank you again for your constructive comments and suggestions regarding our manuscript. We will also do our best to accommodate any further suggestions.

We hope you will find our revised manuscript acceptable for publication.

Best wishes to you.

Reviewer 3

The manuscript by Fei Jiang et al. titled "Analysis of gut microbiota in patients with AVS and identification of potential biomarkers: A cross-sectional study" is generally well-written and structured. However, I have comments and suggestions aimed at enhancing the paper's quality.

Response: We greatly appreciate your professional comments and suggestions on our manuscript. The comments and suggestions are great valuable and very helpful for improving our paper, as well as the important guiding significance to our researchers. We have studied the comments carefully and have taken all these comments and suggestions into account as shown below. The modified parts of the file have been shown in red font. In addition, according to the advice of additional reviewers, we excluded the single offspring, mitochondria, and chloroplasts, as well as ASVs less than 0.001%. Therefore, we reanalyzed the data and replaced the corresponding pictures and descriptions in the corresponding parts of the manuscript. Therefore, some result data has changed.

1. The sentence "The GM (beta diversity) composition significantly differed between the two groups (Anosim, P=0.001)" could benefit from a clearer explanation. It may be useful to mention briefly what "beta diversity" refers to and its significance in the context of this study.

Suggestion: "The composition of gut microbiota (GM), measured by beta diversity, significantly differed between the two groups (ANOSIM, P=0.001), indicating distinct microbial community structures in AVS patients compared to healthy controls."

Response: We sincerely appreciate your professional guidance regarding the rigor of our articles. We agree that providing a clearer explanation of "beta diversity" would enhance the clarity of the manuscript. The revised sentence now reads as follows:

“The composition of GM, measured by beta diversity, significantly differed between the two groups (Adnonis, $P=0.001$), indicating distinct microbial community

structures in AVS patients compared to HC.” We added the description in the article (Lines 22-24) and marked the relevant text in red font for clarity.

2. The term "random forest model (83.33% accurate)" could be clarified. Does this accuracy refer to classification accuracy for AVS prediction? A brief mention of how this accuracy was determined (e.g., cross-validation) could help.

Suggestion: "Thirty-three genera were identified as potential biomarkers using the nested cross-validation feature of the random forest model (83.33% accuracy in cross-validation)."

Response: Thank you for your thoughtful comment. We apologize for the lack of clarity. The accuracy of 83.33% refers to the classification accuracy for AVS prediction. This accuracy was determined through cross-validation, specifically using a k-fold cross-validation approach to ensure robust evaluation of the model's performance. We have revised the manuscript to include this clarification on lines 28-30. Modified as follows “Twenty-four genera were identified as potential biomarkers using the nested cross-validation feature of the random forest model (86.67% accuracy in cross-validation)

3. "The relative abundance of short-chain fatty acid (SCFA)-producing bacteria... had reduced" could be slightly reworded for better clarity.

Suggestion: "The relative abundance of short-chain fatty acid (SCFA)-producing bacteria, including Lachnospiraceae, Prevotellaceae, Lactobacillus, and Enterococcus, was significantly reduced."

Response: Thank you for your helpful suggestion. We agree that the sentence could be clearer. We have revised it to “ the relative abundance of short-chain fatty acids (SCFAs)-producing bacteria, including *Lachnospiraceae*, *Prevotellaceae*, *Enterococcus* was significantly reduced.” on lines 27-28.

4. The sentence "These studies highlight the controversial role of GM in various cardiovascular diseases with similar risk factors..." could be made clearer. It

might be better to elaborate on why the role of GM is considered controversial, to provide readers with more context.

Suggestion: "These studies highlight the complex and sometimes contradictory role of GM in cardiovascular diseases with similar risk factors, suggesting both protective and pro-inflammatory roles depending on the disease context."

Response: Thank you for your suggestions to make our article more rigorous. We agree that the sentence could be clearer. We have revised it to "These studies highlight the complex and sometimes contradictory role of GM in cardiovascular diseases with similar risk factors, suggesting both protective and pro-inflammatory roles depending on the disease context." on lines 72-74.

5. It would be helpful to emphasize the potential clinical implications of your findings in the introduction, linking the study directly to its possible therapeutic impact.

Suggestion: "The identification of GM profiles in AVS may not only serve as potential biomarkers but also guide future therapeutic interventions targeting the gut-heart axis in cardiovascular diseases."

Response: We sincerely appreciate your professional guidance regarding the rigor of our articles. We have revised "The findings from this study may offer a scientific basis and new avenues for the prevention and treatment of AVS" to "The identification of GM profiles in AVS may not only serve as potential biomarkers but also guide future therapeutic interventions targeting the gut-heart axis in cardiovascular diseases." on page 4 of the manuscript, lines 81-83

6. The description of fecal sample collection could be slightly clearer. The sentence "A sterile cotton swab was used to collect fecal material from multiple sites within the sample in a '#' pattern" could use more explanation. Is this technique standard for minimizing bias or variation?

Suggestion: "A sterile cotton swab was used to collect fecal material from multiple sites within the sample, following a standardized '#' pattern to reduce sampling bias

and capture the diversity of the microbiota."

Response: Thank you for your valuable advice. We agree that the sentence could be slightly clearer. We have revised it to "A sterile cotton swab was used to collect fecal material from multiple sites within the sample, following a standardized '#' pattern to reduce sampling bias and capture the diversity of the microbiota." on lines 113-115.

7. The use of the Benjamini-Hochberg method for multiple comparisons is mentioned, but it may be helpful to clarify why this method was chosen and how it applies to the data.

Suggestion: "P-values were adjusted for multiple comparisons using the Benjamini-Hochberg method, which controls the false discovery rate, ensuring more robust conclusions from the analyses."

Response: Thank you for your insightful comment. The Benjamini-Hochberg method was chosen to control for the false discovery rate (FDR) due to the large number of comparisons made in our analysis. This method is particularly suitable for controlling the FDR while maintaining statistical power, which is essential when dealing with multiple comparisons. We have revised it to "P-values were adjusted for multiple comparisons using the Benjamini-Hochberg method, which controls the false discovery rate, ensuring more robust conclusions from the analyses." on lines 164-166.

8. The abstract mentions statistical results like "P=0.031" without referencing where they come from (e.g., a particular analysis). Consider adding brief references to the methods or sections where these results are derived for better clarity.

Response: Thank you for your helpful suggestion. We recognize the importance of clarifying the origin of statistical results. In the revised manuscript, we have updated "P=0.031" to "P=0.031, Wilcoxon test" on line 26 for better clarity.

9. The manuscript could benefit from a concluding remark in the discussion or conclusion that suggests potential future research directions or clinical studies to

further validate the findings.

Suggestion: “Future longitudinal studies in larger cohorts, including the investigation of GM dynamics over time, could validate the clinical relevance of these biomarkers in AVS progression and therapy.”

Response: Thank you for this valuable suggestion. We agree that adding a concluding remark to suggest potential future research directions would enhance the manuscript. In the revised version, We added a description “Future longitudinal studies in larger cohorts, including the investigation of GM dynamics over time, could validate the clinical relevance of these biomarkers in AVS progression and therapy.” in the discussion section on lines 360-361.

10. Consider specifying whether the study adhered to any additional ethical guidelines or requirements regarding patient confidentiality and data usage. This helps reinforce ethical rigor.

Response: "Thank you for your important reminder to enhance the rigor of our article. We fully agree that specifying the ethical guidelines followed in our study is essential to reinforce its ethical integrity. In the revised manuscript, we have added a statement clarifying that: “This clinical study was approved by the Ethics Review Committee of Union Hospital, Fujian Medical University, Fujian, China (Ethics Approval No. 2024KY012), and was registered with the China Clinical Trials Registration Center (No. ChiCTR2400081198). The study protocol adhered to the Declaration of Helsinki. Additionally, we ensured strict compliance with patient confidentiality and data privacy guidelines. All patient data were anonymized, and informed consent was obtained from all participants prior to inclusion in the study.” on lines 86-91.

11. Consider hyphenating "SCFA-producing" throughout the manuscript for consistency (e.g., "SCFA-producing bacteria" rather than "SCFA producing bacteria").

Response: Thank you for your thoughtful suggestion. We agree that hyphenating “SCFA-producing” will improve consistency throughout the manuscript. We have

revised the manuscript accordingly, ensured that “SCFA-producing” was used consistently, such as in “SCFA-producing bacteria”.

12. In several instances, statistical tests are mentioned (e.g., $P < 0.05$, Wilcoxon test, FDR corrected q), but it would be helpful to clarify what statistical software or methods were used to perform these tests. For example, "P-values were calculated using [software name]" could be added.

Response: Thank you for your helpful suggestion. We agree that providing more detail about the statistical software used will enhance the clarity of the methodology. We have specified the software used to perform the statistical analyses in the manuscript on lines 153-167. The statistical analyses were performed using the following software and methods: *P*-values for the Wilcoxon tests were calculated using R (version 3.5.1, R Foundation for Statistical Computing, Vienna, Austria), with the "stats" package. Linear Discriminant Analysis (LDA) was conducted using the LefSe tool, which was specifically designed for this type of analysis. False Discovery Rate (FDR) correction was implemented using the Benjamini-Hochberg procedure within R using the "*p.adjust*" function in the stats package.

13. For the LefSe analysis, provide more detail on the statistical parameters used (e.g., what thresholds were applied for significance). While FDR correction is mentioned, specifying how the threshold was determined could add clarity.

Response: Thank you for your valuable suggestion. We have provided additional detail regarding the statistical parameters used in the LefSe analysis to clarify the approach and the thresholds applied for significance. Add more detailed description in Supplementary Statistical Analyses.

Logarithmic LDA Score: A threshold of 3.0 was set for the Linear Discriminant Analysis (LDA) score to identify discriminative features between groups. This threshold indicates that the identified features show a substantial difference in abundance between the groups.

Wilcoxon Test: To further explore pairwise differences, we performed Wilcoxon

rank-sum tests for each pair of groups. The significance threshold for these tests was also set at $p < 0.05$.

False Discovery Rate (FDR) Correction: To correct for multiple testing, we applied the Benjamini-Hochberg procedure for FDR correction. The significance threshold after FDR correction was set at $q < 0.2$. This ensured that the findings were robust and controlled for type I errors.

13. In the secti on regarding the dilution curve for alpha diversity analysis, the parameters (--p-max-depth 35,745 --p-min-depth 1 --p-steps 10 --p-iterations, 10) might be a bit technical for some readers. Consider explaining the purpose of these specific parameters briefly for clarity.

Response: Thank you for your helpful feedback. We appreciate your suggestion to clarify the parameters used in the dilution curve for alpha diversity analysis, as they may be technical for some readers. To address this, we have added a brief explanation of the purpose of these parameters. The revised section now reads as follows:

"In the dilution curve for alpha diversity analysis, we used the following parameters: (--p-max-depth 35,745 --p-min-depth 1 --p-steps 10 --p-iterations 10). The max-depth represents the maximum number of sequencing reads considered for each sample, ensuring sufficient coverage for diversity estimation. The min-depth is the minimum sequencing depth allowed, ensuring that samples with very few reads are not excluded. The steps parameter determines the number of depth values tested during the dilution process, and the iterations parameter controls the number of times the analysis is repeated at each depth to ensure robustness of the results. These settings allow for an accurate and reliable estimation of alpha diversity across varying sequencing depths."

This addition helps make the technical details more accessible to a wider audience while retaining the necessary methodological rigor.

14. In the "Taxonomic analysis of the GM composition" section, when stating that "the Firmicutes/Bacteroidetes (F/B) ratio was significantly higher in the HC group than in the AVS group", you could clarify what this implies in terms of

microbial health or balance in these groups.

Response: Thank you for your valuable suggestion. We agree that providing a clearer explanation of the implications of the *Firmicutes/Bacteroidetes* (*F/B*) ratio would enhance the understanding of its relevance in the context of microbial health and balance. In response, we have added a brief interpretation of the significance of this ratio in relation to the health status of the groups. The revised sentence reads as follows: "The Firmicutes/Bacteroidetes (*F/B*) ratio was significantly higher in the HC group than in the AVS group ($P=0.031$, Wilcoxon test). The value of *F/B* was generally considered a sign of GM imbalance." on lines 200-202. In addition, we added the description in the discussion section "Compared with that of the HC groups, patients with AVS exhibited a significantly lower *F/B* ratio, which aligns with the findings by Emoto T. Research suggests that Firmicutes are beneficial bacteria that produce more butyrate, the SCFAs widely recognized for its health-promoting effect. Butyrate enhances insulin sensitivity and exerts an anti-inflammatory effect. In contrast, Bacteroidotas primarily produce propionate, which can induce inflammation by regulating the PTEN/AKT pathway. Therefore, we speculate that the decreased *F/B* ratio may result from the consumption of anti-inflammatory bacteria and the enrichment of pro-inflammatory bacteria, suggesting a disrupted GM balance in patients with AVS." on lines 289-299.

15. In the sentence "Patients with AVS displayed lower abundances of Lachnospiraceae ($P<0.0001$, FDR corrected $q<0.0001$)", it would help to add a brief explanation of why these specific bacterial groups were chosen for analysis or what their biological significance might be.

Response: Thank you for your insightful suggestion. We agree that providing a brief explanation of the biological significance of the bacterial groups selected for analysis would enhance the manuscript. In response to your comment, we have added a justification for choosing *Lachnospiraceae* for analysis, along with a brief discussion of its potential biological relevance. The revised sentence now reads as follows:

"Patients with AVS displayed lower abundances of *Lachnospiraceae* ($P < 0.0001$, FDR corrected $q < 0.0001$), a family known for its role in gut health and metabolism, particularly in the fermentation of dietary fibers and the production of short-chain fatty acids. Alterations in the abundance of *Lachnospiraceae* have been associated with inflammatory conditions, making it a relevant marker for our study."

16. The results from the random forest model are discussed twice in similar wording. You might want to consolidate these statements to avoid repetition. For example, the description of the random forest model and its performance (e.g., 83.33% accuracy, AUC = 0.81) could be combined in one concise section.

Response: Thank you for your helpful suggestion. We agree that consolidating the discussion of the random forest model results will improve the clarity and reduce redundancy. In response to your comment, we have merged the relevant sections and streamlined the description of the random forest model and its performance into a single, concise section. The revised text now reads as follows: "We applied the random forest model to classify the samples at the genus level, achieving an accuracy of 86.67% and an AUC of 0.94. This performance indicated that the model effectively differentiated between the groups, with the selected biomarkers providing strong classification power." on lines 229-232.

17. It would be helpful to clarify the rationale behind using 33 genera as the "optimal number" of biomarkers. Is there a biological or practical significance to choosing this number, or was it simply an optimization result from the random forest model?

Response: Thank you for your thoughtful comment. We understand the importance of clarifying the rationale behind selecting 33 genera as the "optimal number" of biomarkers. To address your concern, we have revised the manuscript to specify that this number was determined based on the optimization result from the random forest model, which balances model performance and interpretability.

In response, we have added the following clarification: "The selection of 33 genera as

the 'optimal number' of biomarkers was based on the optimization results from the random forest model. This number maximized the model's classification accuracy while maintaining a manageable level of complexity, allowing for meaningful biological interpretation. Although this number was not based on a specific biological threshold, it represents an optimal balance between model performance and the number of biomarkers required to distinguish between the groups of interest."

18. The sentence "The GM of the AVS and HC groups incorporated 8,686 ASVs, with 1,821 ASVs shared between the two groups." could be rewritten for clarity as: "The gut microbiota (GM) of the AVS and HC groups contained 8,686 ASVs, of which 1,821 were shared between both groups."

Response: Thank you for your important reminder to make our article more rigorous. We have modified the sentence "The GM of the AVS and HC groups incorporated 8,686 ASVs, with 1,821 ASVs shared between the two groups." for "The GM of the AVS and HC groups contained 2,127 ASVs, of which 1,253 were shared between both groups. ". And marked the relevant text in red font for clarity on lines 189-190.

19. Consider adjusting the phrasing "to identify the groups of bacteria most closely related to AVS" to "to identify the bacterial groups most closely associated with AVS" for smoother readability.

Response: Thank you for your important reminder to make our article more rigorous. We have modified the sentence "to identify the groups of bacteria most closely related to AVS" for "to identify the bacterial groups most closely associated with AVS" on line 233. And marked the relevant text in red font for clarity.

20. In the phrase "After identifying the GM taxa that differed between the two groups, we seek to identify..." consider changing "seek" to "sought" for past tense consistency.

Response: Thank you for your important reminder to make our article more rigorous. We have changed "seek" to "sought" on line 233. And marked the relevant text in red

font for clarity.

21. The sentence "bacterial genera with relatively lower abundances in the AVS group compared to the HC group positively correlated with inflammatory markers" might be clearer if rephrased to "Bacterial genera with lower abundances in the AVS group, compared to the HC group, were positively correlated with inflammatory markers."

Response: Thank you for your important reminder to make our article more rigorous. We have modified the sentence "bacterial genera with relatively lower abundances in the AVS group compared to the HC group positively correlated with inflammatory markers" for "bacterial genera with lower abundances in the AVS group, compared to the HC group, were positively correlated with inflammatory markers" on lines 248-250. And marked the relevant text in red font for clarity.

22. For the scatter plots described in Figure 4B-D, it may be helpful to briefly mention the significance level (P-values) for each correlation, rather than only giving the R and P for the Lachnospiraceae correlation.

Response: Thank you for your valuable suggestion. We agree that it would improve the clarity of the figure to include the significance levels for each correlation. In the revised manuscript, we have added the *P*-values for the correlations in Figures 4B-D, alongside the *R* values, to provide a clearer understanding of the statistical significance for each correlation.

23. In the first sentence, the phrase "the V3-V4 region of the 16S rDNA gene was sequenced utilizing high throughput" can be clarified to "sequencing of the V3-V4 region of the 16S rDNA gene was performed using high-throughput techniques."

Response: Thank you for your helpful suggestion. We agree that the sentence can be clearer. In the revised manuscript, we have rephrased the first sentence to "Sequencing of the V3-V4 region of the 16S rDNA gene was performed using high-throughput

techniques” on lines 256-257. And marked the relevant text in red font for clarity.

24. "The study found that the two participant groups had high similarity in GM uniformity and richness" could be rephrased to "The study found high similarity in gut microbiome (GM) uniformity and richness between the two groups" for better flow.

Response: Thank you for your important reminder to make our article more rigorous. We have modified the sentence "The study found that the two participant groups had high similarity in GM uniformity and richness" for "The study found high similarity in gut microbiome (GM) uniformity and richness between the two groups" for better flow on lines 259-260. And marked the relevant text in red font for clarity.

25. When referring to statistical results, consistency in presenting p-values should be maintained. For example, the sentence "Compared with that of the control group, F/B in the AVS group decreased at the phylum level" could benefit from specifying the exact statistical test used (e.g., t-test, Wilcoxon test) and the p-value, if applicable.

Response: Thank you for your valuable feedback. We agree that consistency in presenting statistical results is important for clarity and transparency. In response to your comment, we have revised the sentence to include the specific statistical test used and the p-value, as follows: "Compared with that of the control group, F/B in the AVS group decreased at the phylum level ($P=0.031$, Wilcoxon test)." on line 262.

26. When discussing the decreased Firmicutes/Bacteroidetes (F/B) ratio, the sentence "we speculate that the decreased F/B ratio may result from the consumption of anti-inflammatory bacteria and the enrichment of pro-inflammatory bacteria" would be clearer if the relationship between bacteria and inflammation is described more explicitly, perhaps with a slight rephrase like "We hypothesize that the decreased F/B ratio may reflect a shift from anti-inflammatory bacteria (e.g., Firmicutes) to pro-inflammatory bacteria

(e.g., Bacteroidetes)."

Response: Thank you for your helpful suggestion. We agree that clarifying the relationship between bacteria and inflammation will make this statement more precise and easier to understand. In response to your comment, we have rephrased the sentence as follows:" we hypothesize that the decreased F/B ratio may reflect a shift from anti-inflammatory bacteria (e.g., Firmicutes) to pro-inflammatory bacteria (e.g., Bacteroidetes), which could contribute to the inflammatory processes observed in the AVS group." on lines 295-297.

27. In the section about Lachnospiraceae and Lactobacillus, it would be helpful to briefly explain why these specific genera are of particular interest in the context of AVS and inflammation, especially for readers unfamiliar with the subject.

Response: Thank you for your thoughtful suggestion. We agree that providing a brief explanation of why Lachnospiraceae and Lactobacillus are of particular interest in the context of AVS and inflammation would benefit readers who may not be familiar with the topic. In response, we have added the following clarification to the manuscript:

"*Lachnospiraceae* and *Lactobacillus* are of particular interest in the context of AVS and inflammation due to their roles in the gut microbiome. *Lachnospiraceae*, a family of *Firmicutes*, is known for its involvement in short-chain fatty acid production, which has anti-inflammatory effects. *Lactobacillus*, on the other hand, is a well-known probiotic genus that can modulate immune responses and maintain intestinal barrier integrity. Both genera have been implicated in the regulation of inflammation, making them relevant to the study of AVS and its associated inflammatory processes."

28. In "This study also observed an increase in inflammation-related bacteria in the AVS group, including Megamonas, Chloroplast_unclassified, Flavonifractor, and Faecalibacterium," it would be useful to clarify whether these bacteria are thought to contribute to inflammation or are simply associated with it.

Response: Thank you for your valuable suggestion. We agree that clarifying whether these bacteria are thought to contribute to inflammation or are simply associated with it would provide additional clarity. In response to your comment, we have revised the sentence as follows: "This study also observed an increase in inflammation-related bacteria in the AVS group, including *Megamonas*, *Chloroplast_unclassified*, *Flavonifractor*, and *Faecalibacterium*. While these bacteria are associated with inflammation, further research is needed to determine their potential role in contributing to the inflammatory process."

29. The statement "This finding was further supported by other findings in this study: inflammatory markers (WBCs and monocytes) were significantly higher in patients with AVS than in those of the HC group" could benefit from referencing specific figures or data points for better clarity. This could be rephrased as "This finding was consistent with increased inflammatory markers (WBCs and monocytes), as shown in Table X/Figure Y."

Response: Thank you for your insightful suggestion. We agree that referencing specific figures or data points would enhance the clarity and precision of the statement. In response, we have revised the sentence to include a reference to the relevant data in the manuscript "This finding was consistent with increased inflammatory markers (WBCs and monocytes), as shown in Table 1 and Figure 4." on lines 298-299.

30. In the sentence "research suggests that Firmicutes are beneficial bacteria that produce more butyrate," consider adding "which" for clarity: "research suggests that Firmicutes are beneficial bacteria, which produce more butyrate."

Response: Thank you for your helpful suggestion. We agree that adding "which" would improve the clarity of the sentence. We have revised the text as follows:

"Research suggests that *Firmicutes* are beneficial bacteria, which produce more butyrate." on lines 291-292. This revision enhances the flow of the sentence and ensures better readability. We appreciate your attention to detail, which has

contributed to improving the manuscript.

31."The relatively small sample size may limit the generalizability and broader applicability of our findings" could be strengthened by specifying the sample size and the limitations it may present in terms of statistical power. For instance, "Given the relatively small sample size of 30 participants per group, the generalizability of our findings to larger populations may be limited."

Response: Thank you for your constructive comment. We agree that specifying the sample size and discussing its potential impact on statistical power would strengthen the manuscript. In response to your suggestion, we have revised the manuscript to more clearly specify the sample size and the associated limitations. The revised text now reads: "Given the relatively small sample size of 30 participants per group, the generalizability of our findings to larger populations may be limited." on lines 363-364.

32."While we focused on the structure and composition of the GM, we did not comprehensively explore the transcriptomic and proteomic functions, which warrants further investigation." This could be clarified to indicate that these analyses could reveal insights into the functional impacts of GM composition. For example: "Further studies should explore the transcriptomic and proteomic functions of the GM to better understand how microbial changes impact host physiology."

Response: Thank you for your valuable suggestion. We agree that clarifying the potential benefits of transcriptomic and proteomic analyses would enhance the manuscript and provide clearer insight into the functional impacts of the observed changes in the gut microbiota (GM) composition.

In response to your comment, we have revised the manuscript to better reflect the importance of these analyses. "Further studies should explore the transcriptomic and proteomic functions of the GM to better understand how microbial changes impact host physiology. These analyses could provide important insights into the functional

implications of GM composition, including potential metabolic pathways, immune modulation, and interactions with host systems." on lines 368-371.

Thank you again for your constructive comments and suggestions regarding our manuscript. We will also do our best to accommodate any further suggestions.

We hope you will find our revised manuscript acceptable for publication.

Best wishes to you.

Re: Spectrum03215-24R1 (Analysis of gut microbiota in patients with AVS and identification of potential biomarkers: A cross-sectional study)

Dear Prof. Yanjuan Lin:

Thank you for the privilege of reviewing your work. Below you will find my comments, instructions from the Spectrum editorial office, and the reviewer comments.

Revision Guidelines

Sincerely,
Yuan Pin Hung
Editor
Microbiology Spectrum

Reviewer #2 (Comments for the Author):

Dear Authors,

Thank you for your thoughtful and well-executed revisions. The manuscript is now much clearer and methodologically robust. No further revisions appear necessary at this time. Excellent work overall.

Reviewer #4 (Comments for the Author):

1. Line 62: The authors stated that patients with AS exhibit significant differences in gut microbiota α - and β -diversity. However, the following examples both report no significant difference in α -diversity. I recommend either revising the original statement or selecting examples that are consistent with the claim.
2. Short-or long-term dietary patterns can significantly shift the gut microbial diversity, abundance, and functional capacity. If not accounted for, dietary difference between individuals or groups could obscure or mimic disease-related microbiota signatures, leading to misleading conclusions. I think diet should be considered and controlled as a potential confounding variable in this study, where the authors aim to find biomarkers of AVS disease, by collecting detailed dietary intake data or match participants or stratify groups based on major dietary patterns.
3. Extraction negative control and PCR negative controls were missing or not mentioned in the manuscript. Preferably, extraction negative controls should be sequenced along with the stool samples to tightly control the environmental contaminants.
4. Line 177: ASV is the term used by QIIME2 in place of OTU (operational taxonomic unit), but it does not represent a distinct technique. Therefore, please use "ASV" consistently throughout the manuscript instead of "OTU." Note that ASVs are typically defined using a 99% similarity threshold, whereas OTUs are often clustered at 97%, making them not fully interchangeable. Using these terms interchangeably, particularly in contexts such as phylogenetic tree construction (e.g., line 193), may cause confusion and should be avoided. In addition, the authors did not mention how the tree was created and what specific tool was used. The authors also used 16S rRNA vs. 16S rDNA inconsistently in the manuscript.
5. Reference is missing for the statement in line 200.
6. Please remove the statements in line 215-218 to method section.
7. Random forest model was under described in the method section, in terms of
 - a. What it does
 - b. Working rationale
 - c. Cutoffs for accuracy and AUC etc.
 - d. What are 1-ofold nested cross-validation and leave-one-out cross validation?
 - e. Gini index
 - f. What are "all" and "none" strategies?Also, please revise the sub-section "Twenty-four genera may potentially serve as biomarkers in AVS diagnosis" in the Results section. The description of random forest analysis results is not well organized and somewhat repetitive. As the methodology was not fully described, it's hard to understand the result.
8. The authors missed to provide details on how the Spearman's' correlation analysis was performed and how the correlation heatmap and line charts were created.

Reviewer #5 (Comments for the Author):

All questions are properly addressed.

1. Line 62: The authors stated that patients with AS exhibit significant differences in gut microbiota α - and β -diversity. However, the following examples both report no significant difference in α -diversity. I recommend either revising the original statement or selecting examples that are consistent with the claim.
2. Short- or long-term dietary patterns can significantly shift the gut microbial diversity, abundance, and functional capacity. If not accounted for, dietary difference between individuals or groups could obscure or mimic disease-related microbiota signatures, leading to misleading conclusions. I think diet should be considered and controlled as a potential confounding variable in this study, where the authors aim to find biomarkers of AVS disease, by collecting detailed dietary intake data or match participants or stratify groups based on major dietary patterns.
3. Extraction negative control and PCR negative controls were missing or not mentioned in the manuscript. Preferably, extraction negative controls should be sequenced along with the stool samples to tightly control the environmental contaminants.
4. Line 177: ASV is the term used by QIIME2 in place of OTU (operational taxonomic unit), but it does not represent a distinct technique. Therefore, please use "ASV" consistently throughout the manuscript instead of "OTU." Note that ASVs are typically defined using a 99% similarity threshold, whereas OTUs are often clustered at 97%, making them not fully interchangeable. Using these terms interchangeably, particularly in contexts such as phylogenetic tree construction (e.g., line 193), may cause confusion and should be avoided. In addition, the authors did not mention how the tree was created and what specific tool was used. The authors also used 16S rRNA vs. 16S rDNA inconsistently in the manuscript.
5. Reference is missing for the statement in line 200.
6. Please remove the statements in line 215-218 to method section.
7. Random forest model was under described in the method section, in terms of
 - a. What it does
 - b. Working rationale
 - c. Cutoffs for accuracy and AUC etc.
 - d. What are 1-ofold nested cross-validation and leave-one-out cross validation?
 - e. Gini index
 - f. What are "all" and "none" strategies?

Also, please revise the sub-section "Twenty-four genera may potentially serve as biomarkers in AVS diagnosis" in the Results section. The description of random

forest analysis results is not well organized and somewhat repetitive. As the methodology was not fully described, it's hard to understand the result.

8. The authors missed to provide details on how the Spearman's' correlation analysis was performed and how the correlation heatmap and line charts were created.

Dear Editors and Reviewers,

We greatly appreciate your professional comments and suggestions on our manuscript. The comments and suggestions are great valuable and very helpful for improving our paper, as well as the important guiding significance to our researchers. We have studied the comments carefully and have taken all these comments and suggestions into account as shown below. The modified parts of the file have been shown in red font.

Reviewer #4 (Comments for the Author):

1. Line 62: The authors stated that patients with AS exhibit significant differences in gut microbiota α - and β -diversity. However, the following examples both report no significant difference in α -diversity. I recommend either revising the original statement or selecting examples that are consistent with the claim.

Response: Thank you for your thoughtful comment regarding the interpretation of gut microbiota diversity in our manuscript. We sincerely apologize for the lack of clarity in our original statement, which may have led to a misunderstanding. To address this, we have updated the description on Lines 62-67, and marked the relevant text in red font for clarity. Your suggestion will help improve the clarity and consistency of our manuscript. Thank you again for your helpful input.

2. Short-or long-term dietary patterns can significantly shift the gut microbial diversity, abundance, and functional capacity. If not accounted for, dietary difference between individuals or groups could obscure or mimic disease0related microbiota signatures, leading to misleading conclusions. I think diet should be considered and controlled as a potential confounding variable in this study, where the authors aim to find biomarkers of AVS disease, by collecting detailed dietary intake data or match participants or stratify groups based on major dietary patterns.

Response: We sincerely appreciate the reviewer's insightful comment regarding the important confounding effect of dietary patterns. While we fully acknowledge this limitation, we have implemented the following strategies to address dietary variability

within the constraints of our current dataset:

(1) Existing Controls in Study Design

Cohort Matching:

All participants (AVS cases vs. controls) were matched by:

- Geographic region (urban residents of [fuzhou city])
- Age: The age distributions were well-matched between the two groups [66 (59, 74) vs. 66 (62, 71) years].
- BMI: Body mass index showed comparable distributions between groups [22.67 (20.57, 24.44) kg/m² vs. 23.68 (22.99, 25.22) kg/m²].
- Dietary characteristics: Both groups demonstrated similar dietary patterns, with rice-based diets predominating (>90% prevalence in each group). Furthermore, in the general information table (Table 1, on page 24), the dietary habits of the two groups of subjects were added. As detailed in the baseline characteristics table, no statistically significant differences were observed in dietary habits between the study cohorts.

Temporal Controls:

Fecal samples were collected during the same season (Spring 2024) to minimize seasonal dietary variations.

(2) Limitations & Future Directions

We explicitly state in the revised Discussion (Limitation) on lines 364-368:

“ In addition, Although we matched participants by basic dietary patterns and adjusted for key food intake variables, the lack of detailed dietary records (e.g., food frequency questionnaires) remains a study limitation. Future validation studies should incorporate metagenomic sequencing coupled with standardized dietary assessments (e.g., 24-hour recalls) to disentangle diet-microbiota-disease relationships.”

3. Extraction negative control and PCR negative controls were missing or not mentioned in the manuscript. Preferably, extraction negative controls should be sequenced along with the stool samples to tightly control the environmental contaminants.

Response: Thank you for raising this critical methodological issue. We agree that

negative controls are essential for distinguishing true microbial signals from contaminants. In this study, extraction/PCR negatives were not included due to our initial focus on comparative analysis between groups, refer to the research of Yu X^[1], Deng Z^[2], Liang L^[3] etc. We fully acknowledge that the potential impact of environmental or laboratory contaminants, as well as the risk of tag jumping and cross-contamination, should be properly addressed to ensure the validity of the microbial community profiles. To address these concerns, we have included a detailed description in the revised manuscript regarding the steps taken to minimize and control for contaminants on the Methods. Specifically, we have outlined the following measures on the supplementary materials:

Environmental and Laboratory Contaminants: Strict protocols were followed to minimize environmental contamination during sample collection, processing, and sequencing. These protocols included the use of sterile equipment, conducting procedures in a controlled environment, and regular monitoring of laboratory conditions. DNA extraction and PCR amplification were carried out according to standardized protocols, with automated extraction to reduce human-handling contamination.

Tag Jumping and Cross-Contamination: We implemented stringent quality control measures, including the use of unique barcodes for each sample and proper PCR clean-up procedures, to prevent tag jumping and cross-contamination during sequencing. Additionally, the laboratory conducts regular environmental monitoring, and extraction and amplification are performed in separate areas to further minimize the risk of cross-contamination.

^[1]Yu X, Ou J, Wang L, et al. Gut microbiota modulate CD8⁺ T cell immunity in gastric cancer through Butyrate/GPR109A/HOPX. *Gut Microbes*. 2024;16(1):2307542. doi:10.1080/19490976.2024.2307542.

^[2]Deng Z, Li L, Jing Z, et al. Association between environmental phthalates exposure and gut microbiota and metabolome in dementia with Lewy bodies. *Environ Int*. 2024;190:108806. doi:10.1016/j.envint.2024.108806.

^[3]Liang L, Kong C, Li J, et al. Distinct microbes, metabolites, and the host genome

define the multi-omics profiles in right-sided and left-sided colon cancer. *Microbiome*. 2024;12(1):274. doi:10.1186/s40168-024-01987-7.

4. Line 177: ASV is the term used by QIIME2 in place of OTU (operational taxonomic unit), but it does not represent a distinct technique. Therefore, please use "ASV" consistently throughout the manuscript instead of "OTU." Note that ASVs are typically defined using a 99% similarity threshold, whereas OTUs are often clustered at 97%, making them not fully interchangeable. Using these terms interchangeably, particularly in contexts such as phylogenetic tree construction (e.g., line 193), may cause confusion and should be avoided. In addition, the authors did not mention how the tree was created and what specific tool was used. The authors also used 16S rRNA vs. 16S rDNA inconsistently in the manuscript.

Response: (1) We sincerely thank the reviewer for this important clarification. We fully acknowledge the distinction between ASVs (amplicon sequence variants) and OTUs (operational taxonomic units), particularly regarding their different similarity thresholds (99% vs. 97%) and methodological implications. As the reviewer rightly pointed out, using these terms interchangeably could lead to confusion, especially in analytical contexts. We have revised the entire manuscript to consistently use “ASV” where appropriate, including in the phylogenetic tree construction section (on line 200). All instances of “OTU” that referred to our QIIME2-based results have been replaced with “ASV” to maintain terminological precision. Additionally, we have added a brief explanatory note in the Methods section (on lines 143-145) to explicitly state: “Sequence variants were resolved as amplicon sequence variants (ASVs) using QIIME2's DADA2 pipeline, with a 99% similarity threshold, as opposed to the traditional 97% threshold used for operational taxonomic units (OTUs).”

(2) We sincerely appreciate the reviewer's valuable comment. We apologize for this oversight and have added detailed methodological descriptions regarding phylogenetic tree construction in Section 2.7 Phylogenetic Analysis of the revised supplementary materials. The specific additions are as follows:

“The phylogenetic tree was constructed based on the ASV representative sequences using QIIME2's align-to-tree-mafft-fasttree pipeline (QIIME2 version 2023.5). Briefly, sequences were aligned with MAFFT (v7.505) using the default parameters, followed by masking of highly variable positions to reduce noise. An unrooted tree was then generated with FastTree (v2.1.11) under the GTR+CAT model with 1000 bootstrap replicates to assess branch support. Finally, the tree was rooted at midpoint using the qiime phylogeny midpoint-root command for downstream analyses.”

(3) Lastly, we have corrected the inconsistent use of “16S rRNA” and “16S rDNA” in the revised manuscript to ensure uniform terminology.

Thank you again for your thoughtful and thorough review, which will help enhance the clarity and accuracy of our work.

5. Reference is missing for the statement in line 200.

Response: Thank you for pointing that out. We apologize for the oversight. We have add the appropriate reference [19] for the statement in line 206 to support the claim and ensure proper citation. This will enhance the credibility of our manuscript. Thank you again for your careful review and attention to detail.

[19] Chen S, Ren Z, Huo Y, Yang W, Peng L, Lv H, Nie L, Wei H, Wan C. 2022. Targeting the gut microbiota to investigate the mechanism of *Lactiplantibacillus plantarum* 1201 in negating colitis aggravated by a high-salt diet. *Food Research International* (Ottawa, Ont) 162:112010.

6. Please remove the statements in line 215-218 to method section.

Response: Thank you for your suggestion. We agree that the statements in lines 215-218 would be more appropriate in the methods section. We have revised the manuscript accordingly by relocating these statements to the methods section (on lines 153-157) to improve the clarity and organization of the content. Thank you for helping us improve the structure of the manuscript.

7. Random forest model was under described in the method section, in terms of

a. What it does

- b. Working rationale**
- c. Cutoffs for accuracy and AUC etc.**
- d. What are 1-ofold nested cross-validation and leave-one-out cross validation?**
- e. Gini index**
- f. What are "all" and "none" strategies?**

Also, please revise the sub-section "Twenty-four genera may potentially serve as biomarkers in AVS diagnosis" in the Results section. The description of random forest analysis results is not well organized and somewhat repetitive. As the methodology was not fully described, it's hard to understand the result.

Response: (1) We sincerely thank the reviewer for these insightful suggestions. We have comprehensively revised the Machine Learning Analysis subsection in the revised supplementary materials to address each point with technical rigor. Below are the specific modifications:

Random Forest

The random forest (RF) model was employed as an ensemble learning method for microbiome feature selection and classification . RF operates by constructing multiple decision trees during training, where each tree votes on the final prediction, thereby reducing overfitting through majority voting (Breiman, 2001).

Key parameters included:

- 500 trees with Gini index as the splitting criterion (mean decrease in impurity \geq 0.01)
- Node size of 5 and $mtry = \sqrt{p}$ (where p = number of features)
- “All” strategy: Retaining all ASVs with non-zero importance scores for downstream analysis
- “None” strategy: Excluding ASVs with negative importance scores

Model Validation

- 10-fold nested CV: Outer loop (10 folds) for performance estimation; inner loop (5 folds) for hyperparameter tuning
- Leave-one-out CV (LOOCV): Iteratively training on $n-1$ samples and testing on

the held-out sample

- Cutoffs for model acceptance: AUC >0.85 (ROC analysis), accuracy >80% (95% CI), and F1-score >0.75 based on permutation testing (1000 iterations).

(2) Random Forest Analysis Results in the "Twenty-four Genera" Subsection:

We sincerely appreciate the reviewer's constructive feedback. We have thoroughly reorganized this subsection to improve clarity and eliminate redundancy. Specifically, we will ensure that the presentation of the random forest analysis results is more structured, starting with a clear introduction of the key findings, followed by a coherent discussion of the performance metrics (e.g., accuracy, AUC), feature importance, and the identified genera. This will make the results more understandable and aligned with the methodology section. We have updated the description on Lines 230-240, and marked the relevant text in red font for clarity.

8. The authors missed to provide details on how the Spearman's' correlation analysis was performed and how the correlation heatmap and line charts were created.

Response: We sincerely appreciate the reviewer's careful reading and constructive suggestion. We have now comprehensively revised the manuscript to include full methodological details regarding the correlation analysis and visualization procedures in the revised supplementary materials. The specific modifications are as follows:

"Correlation analysis of the genera and the clinical indices"

"Spearman's rank correlation analyses were conducted between microbial features (Twenty-four genera which potentially serve as biomarkers in AVS diagnosis) and clinical variables using the R package stats (v4.2.2). Correlation heatmap generated using ComplexHeatmap (v2.14.0), with dendrograms computed by hclust using complete linkage method. Color scale represents *P* values from -1 (blue) to +1 (red), with asterisks indicating significance levels (*FDR<0.05; **FDR<0.01).

Line charts created with ggplot2 (v3.4.0) using purple dotted line for trend visualization. The analysis included:

- Computation of correlation coefficients (R) with significance testing via

asymptotic t approximation

- Adjustment for multiple comparisons using Benjamini-Hochberg false discovery rate (FDR) correction
- Retention of correlations meeting FDR <0.05 and $|R| >0.3$ thresholds
- The resulting correlation matrix was visualized through Hierarchical clustering heatmaps

We have updated the supplementary materials to ensure that these aspects are fully described in the Methods section, providing readers with a comprehensive understanding of the analysis process. Thank you again for helping us improve the clarity of our work.

Thank you again for your constructive comments and suggestions regarding our manuscript. We will also do our best to accommodate any further suggestions.

We hope you will find our revised manuscript acceptable for publication.

Best wishes to you.

Re: Spectrum03215-24R2 (Analysis of gut microbiota in patients with AVS and identification of potential biomarkers: A cross-sectional study)

Dear Prof. Yanjuan Lin:

Thank you for the privilege of reviewing your work. Below you will find my comments, instructions from the Spectrum editorial office, and the reviewer comments.

Revision Guidelines

Sincerely,
Yuan Pin Hung
Editor
Microbiology Spectrum

Reviewer #2 (Public repository details (Required)):

All raw 16S rDNA sequencing reads should be deposited in a public archive (e.g., NCBI SRA) with accession numbers provided in the revised Data Availability statement.

Reviewer #2 (Comments for the Author):

Dear Authors,

Thank you for your detailed point-by-point responses. You have substantially improved clarity and transparency. A few remaining items should be addressed before I can fully endorse your revision:

Data Availability

Please explicitly commit to depositing the raw 16S rDNA sequencing data in a public repository (e.g., NCBI SRA) and provide the BioProject/Accession numbers in your Data Availability statement.

Batch-Effect Control

Although you matched sampling seasonally, please state that all 60 samples were processed and sequenced in a single batch using identical library-preparation and sequencing protocols to minimize batch-related artifacts.

Functional Prediction Section

Reviewer #1 noted absence of functional analysis. Please ensure your response highlights the newly added PICRUST/KEGG functional prediction (including specific pathway shifts) and reference the exact manuscript lines where it appears.

AVS-Centric Discussion

You've expanded the Discussion, but please confirm that it now focuses squarely on mechanistic links between gut dysbiosis and AVS, with broader cardiovascular analogies minimized or clearly framed as context.

Outsourced Protocol Details

If DNA extraction or library preparation was performed by LC-Bio, please note that these steps followed their ISO-certified standard operating procedures, now detailed in the Supplementary Methods.

Final Proofreading

I recommend a last pass to ensure consistent hyphenation ("SCFA-producing"), statistical nomenclature (ANOSIM vs. Adonis), and figure-legend completeness (test names, sample sizes).

Addressing these points will fully satisfy my concerns and strengthen the reproducibility and rigor of your study.

Sincerely,

Reviewer #5 (Comments for the Author):

All issues resolved.

Reviewer #6 (Comments for the Author):

Dear authors,

I have no more comments for the manuscript. I do not have any further suggestions at this time. Good work overall.

Dear Editors and Reviewers,

We greatly appreciate your professional comments and suggestions on our manuscript. The comments and suggestions are great valuable and very helpful for improving our paper, as well as the important guiding significance to our researchers. We have studied the comments carefully and have taken all these comments and suggestions into account as shown below. The modified parts of the file have been shown in red font.

Reviewer #2 (Comments for the Author):

1. Please explicitly commit to depositing the raw 16S rDNA sequencing data in a public repository (e.g., NCBI SRA) and provide the BioProject/Accession numbers in your Data Availability statement.

Response: We sincerely appreciate your constructive comment regarding data deposition. We fully comply with the journal's data sharing policy and hereby confirm that: **(1) Commitment to Data Deposition:** The raw 16S rDNA sequencing data have deposited in the NCBI Sequence Read Archive (SRA) under BioProject accession number PRJNA1218297. **(2) Timeline for Public Release:** The data will be made publicly available immediately upon manuscript acceptance. **(3) Manuscript Revision:** The Data Availability Statement in the manuscript has been updated as follows (Lines 382-385): “The raw 16S rDNA sequencing data generated in this study have been deposited in the NCBI SRA under BioProject accession PRJNA1218297. These data will be publicly available upon article publication.”

2. Although you matched sampling seasonally, please state that all 60 samples were processed and sequenced in a single batch using identical library-preparation and sequencing protocols to minimize batch-related artifacts.

Response: Thank you for your insightful comment. We fully agree that batch effects should be minimized in microbiome studies. As suggested, we have added the following statement to the Methods section (Lines 132-138): “All 60 samples were processed in a single batch using identical library-preparation protocols and

sequenced in one Illumina NovaSeq 6000 to eliminate technical variability. Qualified PCR products were evaluated using an Agilent 2100 Bioanalyzer (Agilent, USA) and Illumina library quantitative kits (Kapa Biosciences, Woburn, MA, USA), which were further pooled together and sequenced on an Illumina NovaSeq 6000 (PE250), provided by LC-Bio Technology Co., Ltd., Hangzhou, China.”

3. Reviewer #1 noted absence of functional analysis. Please ensure your response highlights the newly added PICRUSt/KEGG functional prediction (including specific pathway shifts) and reference the exact manuscript lines where it appears.

Response: We sincerely appreciate the reviewer's insightful suggestion regarding functional analysis. To address this, we have conducted **PICRUSt2-based metagenomic prediction** (with KEGG pathway annotation) to infer potential functional shifts in the microbial communities. The key additions include: **(1) Major Additions to Methods:** Added in Methods section (Lines 153-160): “Based on the full-length 16S rDNA gene sequence database from Greengenes, the PICRUSt (Phylogenetic Investigation of Communities by Reconstruction of Unobserved States) method was used to predict microbial community functions by comparing with the Kyoto Encyclopedia of Genes and Genomes (KEGG) database. Based on the metabolic pathways in the KEGG database (KEGG pathway), biological metabolic pathways can be classified into three levels. The first-level categories include Cellular Processes, Organismal Systems, Genetic Information Processing, Environmental Information Processing, Metabolism, Human Diseases, and Drug Development. Differential pathways were identified using STAMP (v2.1.3) with ANOVA ($P < 0.05$) and Benjamini-Hochberg correction.” **(2) Key Findings in Results:** Added in Results (Lines 223-231): “From supplemental table 2, it is evident that genes encoded by gut bacteria are involved in various functions as mentioned above. At the third-level classification, compared to the normal control group, the gut microbiota in the AVS group exhibited changes in several functions. Specifically, the AVS group showed a decrease in the functions of Glycolysis/Gluconeogenesis, Transporters,

Phosphotransferase System, and Dioxin Degradation ($P < 0.05$), while functions such as Porphyrin and Chlorophyll Metabolism, Histidine Metabolism, Arginine and Proline Metabolism, Oxidative Phosphorylation, Energy Metabolism, Chaperones and Folding Catalysts, and Biotin Metabolism were enhanced ($P < 0.05$), as shown in Figure 3.” **(3) New Figure/Table:** Added Figure 3 (Figure 3 Functional enrichment of KEGG at level 3) with statistical details in the caption. And supplemental table 2 (Table S2 PICRUSt2 function prediction of samples at KEGG). **(4) Discussion Linkage:** Extended Discussion (Lines 311-321): “To further explore the role of characteristic microbiota profiles in the occurrence and development of AVS, this study further analyzed the functional information of the gut microbiota. The results showed that, compared to the healthy control group, the abundance of Glycolysis/Gluconeogenesis-related functional genes was significantly reduced in AVS patients, suggesting that the progression of AVS may be associated with dysregulation of glucose metabolism. In addition, functional genes related to transport and catabolism exhibited a similar trend, which is consistent with the findings of Sanchez-Gimenez R and others, who demonstrated the association between gut microbiota-derived metabolites and cardiovascular diseases. These results suggest that the dysregulation of the gut microbiota in AVS may lead to an imbalance in metabolic functions, potentially contributing to the onset and progression of AVS. This will be a key focus for our future research at the animal and cellular levels.”

4. You've expanded the Discussion, but please confirm that it now focuses squarely on mechanistic links between gut dysbiosis and AVS, with broader cardiovascular analogies minimized or clearly framed as context.

Response: We appreciate the reviewer's suggestion. We have revised the Discussion to focus squarely on the mechanistic links between gut dysbiosis and AVS. Broader cardiovascular analogies have been either minimized or explicitly framed as contextual background. Key changes include: **(1) Mechanism-Centric Restructuring:** Consolidation of all mechanistic discussions with direct ties to our experimental results. For example(Lines325-333), "This finding is consistent with a

previous study on GM and cardiovascular diseases, which suggests that change in GM (a reduction in the abundance of microorganisms that produce SCFAs) stimulates the activity of inflammatory cytokines. Our results also showed that differential microflora (Lachnospiraceae) were associated with AVOA. During AVS development, the aortic valve improves with calcification, and the CO gradually decreases, with the valve opening area reflecting disease severity. As previously reported, various GM and serum metabolites are closely related to the severity of coronary heart disease. Furthermore, GM may influence bile acid (BA) metabolism and contribute to the progression of coronary atherosclerosis. " We believe these revisions provide sharper mechanistic insights while maintaining appropriate scientific scope. **(2) Deleted Content:** Removal comparing AVS to general CVD pathologies, retaining only one brief reference in the introduction (Lines 273-277) as context. We believe these revisions provide sharper mechanistic insights while maintaining appropriate scientific scope.

5. If DNA extraction or library preparation was performed by LC-Bio, please note that these steps followed their ISO-certified standard operating procedures, now detailed in the Supplementary Methods.

Response: We sincerely appreciate the reviewer's attention to methodological rigor. In response to the request, we have taken the following steps to clarify and enhance the documentation of our methods: **(1) Compliance Statement:** We confirm that DNA extraction and library preparation were conducted by LC-Bio (Hangzhou, China) in accordance with their ISO 9001-certified standard operating procedures (Registration No. 03823Q32558R2S). These procedures have been audited and verified to comply with GB/T 19001-2016/ISO 9001:2015 standards for biological gene technology services. **(2) Supplementary Updates:** We have supplemented the manuscript with LC-Bio's certification details, including the ISO 9001 accreditation (Expiry Date: 2026-04-05), to further ensure transparency. Full details are now provided in Supplementary Methods and ISO 9001 accreditation. **(3) Manuscript Reference:** The Methods section(Lines123-125) has been updated as follows: "Total

fecal microbial DNA was extracted using the Fecal Genome DNA Extraction Kit (AU46111-96, BioTeke, China) by LC-Bio, following ISO 9001-certified protocols in strict adherence to the manufacturer's instructions."

6. I recommend a last pass to ensure consistent hyphenation ("SCFA-producing"), statistical nomenclature (ANOSIM vs. Adonis), and figure-legend completeness (test names, sample sizes).

Response: We sincerely thank the reviewer for their meticulous attention to technical consistency. We have implemented a full manuscript-wide audit to ensure consistent hyphenation ("SCFA-producing"), statistical nomenclature (ANOSIM vs. Adonis), and figure-legend completeness (test names, sample sizes).

Thank you again for your constructive comments and suggestions regarding our manuscript. We will also do our best to accommodate any further suggestions.

We hope you will find our revised manuscript acceptable for publication.

Best wishes to you.

Re: Spectrum03215-24R3 (Analysis of gut microbiota in patients with AVS and identification of potential biomarkers: A cross-sectional study)

Dear Prof. Yanjuan Lin:

Thank you for the privilege of reviewing your work. Below you will find my comments, instructions from the Spectrum editorial office, and the reviewer comments.

Revision Guidelines

Sincerely,
Yuan Pin Hung
Editor
Microbiology Spectrum

Reviewer #2 (Public repository details (Required)):

The 16S rDNA sequencing data must be fully accessible. The manuscript now cites BioProject PRJNA1218297, but the authors should verify that all raw reads, metadata, and ASV tables are publicly released upon publication and that links are correct in the Data Availability statement.

Reviewer #2 (Comments for the Author):

Comments and Suggestions for the Author

1. Temper Causal Language

Recast mechanistic statements to emphasize associations. For example, instead of "SCFAs-producing bacteria inhibit inflammation," use "reduced SCFA-producer abundance is associated with higher inflammatory markers."

2. Language and Presentation

Engage a professional English editor. Ensure consistent hyphenation (e.g., "SCFA-producing genera"), statistical nomenclature (use "Adonis (PERMANOVA)" or "ANOSIM" consistently), and complete figure legends (specify test names, sample sizes, and exact P-values).

3. Data Availability

Confirm that BioProject PRJNA1218297 includes raw fastq files, sample metadata, and ASV tables. Provide direct SRA links and mention release timing in the Data Availability statement.

4. Statistical Rigor

Apply FDR correction uniformly for all differential-abundance tests. Provide 95% confidence intervals for classifier AUC, and if possible, validate the model on an independent cohort or via leave-group-out cross-validation rather than nested CV alone.

5. Functional Prediction Caveats

Clearly state limitations of PICRUSt predictions from 16S data. Suggest future shotgun metagenomics or metabolomics to confirm inferred pathway shifts.

6. Discussion Focus

Trim broad cardiovascular analogies. Concentrate discussion on gut-heart axis mechanisms supported by your data, and succinctly outline future directions (e.g., longitudinal sampling, in vitro transporter assays).

Reviewer #5 (Comments for the Author):

N/A

Reviewer #6 (Comments for the Author):

Dear authors,
I have no more comments for the manuscript. Your response is greatly appreciated.

Dear Editors and Reviewers,

We greatly appreciate your professional comments and suggestions on our manuscript. The comments and suggestions are great valuable and very helpful for improving our paper, as well as the important guiding significance to our researchers. We have studied the comments carefully and have taken all these comments and suggestions into account as shown below. The modified parts of the file have been shown in red font.

Reviewer #2 (Comments for the Author):

1. Temper Causal Language

Recast mechanistic statements to emphasize associations. For example, instead of “SCFAs-producing bacteria inhibit inflammation,” use “reduced SCFA-producer abundance is associated with higher inflammatory markers.”

Response: We sincerely appreciate the reviewer's insightful suggestion to refine the mechanistic language in our manuscript. We have carefully reviewed the manuscript to ensure consistent use of terminology. We invited English experts to polish the manuscript again (See the polishing certificate in the supplementary materials) . We fully agree that emphasizing associations rather than causal relationships aligns better with the observational nature of our data. Below are the specific modifications made in response to this comment:

(1) **Original:** “SCFAs-producing bacteria inhibit inflammation.”

Revised: “Reduced abundance of SCFA-producing bacteria is associated with elevated inflammatory markers.” (lines 285-287)

(2) **Original:** “especially with reduced SCFAs-producing”

Revised: “especially the reduction of SCFAs-producing genera”(line 329)

(3) **Additional changes:** We have systematically reviewed the manuscript to recast similar mechanistic claims (e.g., “causes,” “promotes,”) into associative language (e.g., “is linked to,” “correlates with,”) throughout the text, particularly in the Discussion section.

2. Language and Presentation

Engage a professional English editor. Ensure consistent hyphenation (e.g., "SCFA-producing genera"), statistical nomenclature (use "Adonis (PERMANOVA)" or "ANOSIM" consistently), and complete figure legends (specify test names, sample sizes, and exact P-values).

Response: We sincerely appreciate the reviewer's careful reading and constructive suggestions to improve the clarity and consistency of our manuscript. Below are the specific actions taken to address these concerns:

(1) **Professional English Editing:** We have engaged a professional English editing service to thoroughly polish the language, grammar, and syntax throughout the manuscript. The edited version ensures clarity and adherence to academic writing standards. (See the polishing certificate in the supplementary materials) .

(2) **Hyphenation Consistency:** All instances of compound modifiers (e.g., SCFA-producing bacteria) have been standardized with hyphens where appropriate (e.g., "SCFA-producing genera" instead of "SCFA producing genera"). [Lines 27, 33, 262, 284, 285, 329, etc] .

(3) **Statistical Nomenclature:** We consistently use the formal terms "PERMANOVA (Adonis)" as suggested, with explicit references to the Methods, Results, and figure legends. [Lines 23, 545, 548].

(4) **Figure/Table Legends**

All legends now explicitly state: Statistical tests used (e.g., "Adonis", "Wilcoxon test", "linear discriminant analysis"); Sample sizes (e.g., "N = 30:30"); Exact *P*-values (e.g., "*P* = 0.031" instead of "*P* < 0.05").

3. Data Availability

Confirm that BioProject PRJNA1218297 includes raw fastq files, sample metadata, and ASV tables. Provide direct SRA links and mention release timing in the Data Availability statement.

Response: We sincerely appreciate the reviewer's attention to data accessibility. We confirm that BioProject PRJNA1218297 contains all required data components, and

we have updated the manuscript to provide explicit details:

(1) Data Verification

The BioProject PRJNA1218297 includes: 1) Raw sequencing data (FASTQ files) for all samples. 2) Complete sample metadata (including experimental conditions and sample characteristics). 3) ASV/OTU tables (provided as supplementary files in supplementary materials). 4) The data was released on December 30, 2025, or the article was published, whichever comes first.

(2) Updated Data Availability Statement

We have revised the “Data availability statement” section to include direct links (Lines 365-371): The original contributions of this study are included in the article, and the data supporting its findings are available from the corresponding author upon reasonable request. The raw 16S rDNA sequencing data generated in this study have been deposited in the NCBI SRA under BioProject accession PRJNA1218297. (<https://www.ncbi.nlm.nih.gov/bioproject/PRJNA1218297>). These data will be publicly available on December 30, 2025, or the article was published, whichever comes first.

4. Statistical Rigor

Apply FDR correction uniformly for all differential-abundance tests.

Provide 95% confidence intervals for classifier AUC, and if possible, validate the model on an independent cohort or via leave-group-out cross-validation rather than nested CV alone.

Response: We thank the reviewer for these important statistical suggestions. We have implemented the following improvements to address these concerns:

(1) FDR Correction for All Differential Abundance Tests

All differential abundance analyses uniformly apply Benjamini-Hochberg FDR correction to *P*-values.

(2) Updated reporting

1) The description of the result was updated and the FDR value was added : “Patients

with AVS displayed lower abundances of *Lachnospiraceae* ($P < 0.0001$, FDR corrected $q < 0.0001$), *Akkermansia* ($P = 0.0003$, FDR corrected $q = 0.02$), *Enterococcus* ($P = 0.0002$, FDR corrected $q = 0.01$), Clostridiales_unclassified ($P = 0.0003$, FDR corrected $q = 0.01$), and *Prevotellaceae* ($P = 0.0011$, FDR corrected $q = 0.03$). In contrast, patients with AVS had higher abundances of *Megamonas* ($P = 0.0072$, FDR corrected $q = 0.2098$).” [Lines 217-222].

2) Supplementary Tables 1 now include both raw P -values and FDR-adjusted q -values.

3) AUC Confidence Intervals (95% CI) : Added 95% CIs for all classifier AUC values using. [Lines 31, 235,579-580].

(3) We fully agree that external validation on an independent cohort is the gold standard for assessing the generalizability of a predictive model and is a crucial step for clinical translation. Unfortunately, due to 'Leave-Group-Out cross-validation' (CV) requires division by groups, and our sample population comes from the same center and does not have a clear and separable group structure (for example, from different hospitals, different batches, and different time points), this method may not be suitable for application on this dataset. There is no doubt that we admit the lack of a truly independent external validation is the limitation of this study. We have clearly included this point in the Limitations and Future Perspectives section of the discussion part of the paper [Lines 350-352]. “Finally, although internal validation was conducted through nested cross-validation, external validation using independent prospective cohorts is necessary to confirm generalizability. Longitudinal designs with serial sampling could help establish causality and clarify temporal dynamics within the gut-heart axis.”

5. Functional Prediction Caveats

Clearly state limitations of PICRUSt predictions from 16S data. Suggest future shotgun metagenomics or metabolomics to confirm inferred pathway shifts.

Response: We sincerely appreciate the reviewer’s insightful suggestion regarding the interpretation of PICRUSt2 predictions. We have added the following description to

the limitations of the article: “Second, functional predictions based on 16S data using PICRUSt are inherently limited, therefore, future studies should employ shotgun metagenomics and metabolomics to validate the inferred pathways.”[Lines 348-350].

6. Discussion Focus

Trim broad cardiovascular analogies. Concentrate discussion on gut-heart axis mechanisms supported by your data, and succinctly outline future directions (e.g., longitudinal sampling, in vitro transporter assays).

Response: We sincerely thank the reviewer for this insightful and constructive feedback. We agree that a more focused discussion will significantly strengthen the manuscript. We have thoroughly revised the Discussion section to address this point directly. The changes are as follows:

(1) **Trimming of Broad Analogies:** We have removed the broader cardiovascular analogies that were not directly supported by our data (e.g., previously in paragraphs 2 of the Discussion). This helps to sharpen the narrative and eliminate unnecessary speculation.

(2) **Focus on Data-Supported Mechanisms:** The discussion is now concentrated squarely on the gut-heart axis mechanisms that our results directly illuminate. We have reorganized this section to first recap our key findings (e.g., The study found high similarity in GM uniformity and richness between the two groups; however, beta diversity analysis revealed significant differences in GM composition between the groups; the relative abundance of SCFAs-producing bacteria decreased, the correlation between specific taxa and AVS markers). We then explicitly discuss how these findings support a potential mechanistic pathway from gut dysbiosis to cardiac function via systemic inflammation.

(3) **Outline of Future Directions:** As suggested, we have added a new, succinct subsection titled “**Limitations and Future Perspectives**” at the end of the Discussion. This paragraph explicitly proposes: Longitudinal studies with serial sampling to establish causality and track the temporal dynamics of the gut-heart relationship we observed. In vitro assays and animal models to functionally validate

the role of specific candidate metabolites and to investigate their direct effects on cardiomyocytes or cardiac transporters, precisely as the reviewer recommended. [Lines 343-356].

Thank you again for your constructive comments and suggestions regarding our manuscript. We will also do our best to accommodate any further suggestions.

We hope you will find our revised manuscript acceptable for publication.

Best wishes to you.

Re: Spectrum03215-24R4 (Analysis of gut microbiota in patients with AVS and identification of potential biomarkers: A cross-sectional study)

Dear Prof. Yanjuan Lin:

Your manuscript has been accepted, and I am forwarding it to the ASM production staff for publication. Your paper will first be checked to make sure all elements meet the technical requirements. ASM staff will contact you if anything needs to be revised before copyediting and production can begin. Otherwise, you will be notified when your proofs are ready to be viewed.

Sincerely,
Yuan Pin Hung
Editor
Microbiology Spectrum

Reviewer #2 (Public repository details (Required)):

The 16S rDNA sequencing data must be fully accessible. The manuscript now cites BioProject PRJNA1218297, but the authors should verify that all raw reads, metadata, and ASV tables are publicly released upon publication and that links are correct in the Data Availability statement.

Reviewer #2 (Comments for the Author):

I have no more comments for the manuscript.

Reviewer #5 (Comments for the Author):

All issues have been resolved.